# Label-free imaging of M1 and M2 macrophage phenotypes in the human dermis in vivo using two-photon excited FLIM

**Marius Kröger[1], Jörg Scheffel[1], Evgeny A Shirshin[2], Johannes Schleusener[1], Martina C Meinke[1], Jürgen Lademann[1], Marcus Maurer[1], Maxim E Darvin[1]***

[1]Charité – Universitätsmedizin Berlin, corporate member of Freie Universität Berlin, Humboldt- Universität zu Berlin, and Berlin Institute of Health, Department of Dermatology, Venerology and Allergology, Berlin, Germany; [2]Lomonosov Moscow State University, Faculty of Physics, Moscow, Russian Federation

**Abstract** Macrophages ($M\Phi$s) are important immune effector cells that promote (M1 $M\Phi$s) or inhibit (M2 $M\Phi$s) inflammation and are involved in numerous physiological and pathogenic immune responses. Their precise role and relevance, however, are not fully understood for lack of noninvasive quantification methods. Here, we show that two-photon excited fluorescence lifetime imaging (TPE-FLIM), a label-free noninvasive method, can visualize $M\Phi$s in the human dermis in vivo. We demonstrate in vitro that human dermal $M\Phi$s exhibit specific TPE-FLIM properties that distinguish them from the main components of the extracellular matrix and other dermal cells. We visualized $M\Phi$s, their phenotypes and phagocytosis in the skin of healthy individuals in vivo using TPE-FLIM. Additionally, machine learning identified M1 and M2 $M\Phi$s with a sensitivity of 0.88±0.04 and 0.82±0.03 and a specificity of 0.89±0.03 and 0.90±0.03, respectively. In clinical research, TPE-FLIM can advance the understanding of the role of $M\Phi$s in health and disease.

**\*For correspondence:**
maxim.darvin@protonmail.com

**Competing interest:** The authors declare that no competing interests exist.

## Editor's evaluation

The authors have used measurements of endogenous fluorescence lifetimes in the two-photon stimulated NAD(P)H excitation-emission range to build an in vivo classifying for macrophage differentiation status in human dermis. The training data was derived from in vitro and ex vivo analysis of M1 and M2 polarised macrophages from peripheral blood, isolated from tissue or studies ex vivo in frozen sections with marker based validation. A machine learning approach for in vivo classification is presented and an approach to detect phagocytes in vivo is suggested.

## Introduction

Macrophages ($M\Phi$s) are important immune effector cells in organs and tissues that act as border junctions to environments such as the gut, the airways, and the skin (*Elhelu, 1983*). Skin $M\Phi$s (*Dong et al., 2016*; *Estandarte et al., 2016*; *Ryter, 1985*) originate from circulating monocytes (*Geissmann et al., 2010*; *Gordon and Taylor, 2005*) via the same infiltration route into the dermis as monocyte-derived dendritic cells (*Schmid and Harris, 2014*; *Figure 1a*) and are mainly located in the papillary and reticular dermis in close proximity to blood vessels (*Weber-Matthiesen and Sterry, 1990*; *Figure 1b–d*). It has been known for more than 30 years that skin $M\Phi$s are abundant and heterogeneous, based on their morphology, localization, and staining properties (*Weber-Matthiesen and*

**Figure 1.** Dermal monocyte skin infiltration and CD68 stained M1 and CD163 stained M2 MΦs distribution in excised human skin. Schematic illustration of monocyte (MO) (green) infiltration into tissues and macrophage (MΦ)-polarization into M1 MΦs (yellow) via IFN-γ, LPS, and TNF and M2 MΦs (blue) via IL-4, IL-13, and IL-33 (**a**). Schematic of skin with exemplary locations of monocytes (green), M1 MΦs (yellow), and M2 MΦs (blue) (**b**). Density of M1 MΦs (marked with arrows) stained with CD68 (**c**) and M2 MΦs (marked with arrows) stained with CD163 (**d**) in 10 μm thick cryo-section. Scale bar: 100 μm. The terms M1 and M2 MΦs are simplistic, as many signals modulate MΦ functions, resulting in a spectrum between the M1 and M2 MΦ phenotypes.

*Sterry, 1990*). More recently, skin MΦs have been classified based on their function, and they fall into two phenotypes referred to as inflammation-promoting M1-polarised MΦs (classically activated) and anti-inflammatory M2-polarised MΦs (alternatively activated) (*Arango Duque and Descoteaux, 2014*; *Figure 1a*). M1 MΦs are activated by viral and bacterial infection (*Benoit et al., 2008*; *Ferrer et al., 2019*; *Malmgaard et al., 2004*), interferon-γ, lipopolysaccharide (LPS), and tumor necrosis factor (TNF), which is known as the classical activation pathway (*Li and Liu, 2018*). M2 MΦs are alternatively activated in response to IL-4, IL-13, and IL-33 (*Furukawa et al., 2017*; *Sica and Mantovani, 2012*). Recently, this paradigm was questioned, as a manifold of cytokines, biomarkers, and activators are involved in MΦs functioning, resulting in a continuum of states between the M1 and M2 phenotypes (*Mendoza-Coronel and Ortega, 2017*; *Murray et al., 2014*). Furthermore, different markers like CXCL10 for M1 MΦs and CCL17 for M2 MΦs can have the function to attract t-cells. Macrophages in disease, cancer, or obesity can switch function from wound healing to inflammatory MΦs

given the right signals and microenvironment (*Mosser and Edwards, 2008*). However, for simplicity, the terms M1 and M2 $\mathrm{M\Phi}$s are used here with the activators in brackets, where applicable.

Skin M1 $\mathrm{M\Phi}$s are held to contribute to dermal innate immunity and homeostasis. This is supported by reports that M1 $\mathrm{M\Phi}$s can phagocyte objects up to 20 µm in size (*Morhenn et al., 2002*), promote skin inflammatory and immune responses (*Remmerie and Scott, 2018*; *Theret et al., 2019*; *Yanez et al., 2017*), and produce nitric oxide and other reactive oxygen species (ROS) (*Forman and Torres, 2001*; *Rendra et al., 2019*). Skin M2 $\mathrm{M\Phi}$s, on the other hand, are thought to promote dermal repair, healing, and regeneration, for example, by contributing to the formation of the extracellular matrix (ECM; *Ploeger et al., 2013*).

The precise role of skin $\mathrm{M\Phi}$s and their M1 and M2 phenotypes in health and disease remain to be elucidated. In skin diseases, such as melanoma (*Bardi et al., 2018*), systemic sclerosis (*Trombetta et al., 2018*), Lupus (*Chong et al., 2015*), and LPS tolerance (*O'Carroll et al., 2014*), polarization of $\mathrm{M\Phi}$s leading to mixed M1 and M2 phenotypes can be observed. It is not known whether and in what density mixed $\mathrm{M\Phi}$ phenotypes are to be expected in healthy skin. The fluorescence properties of mixed phenotypes have not been studied.

Efforts to do so include their quantification in healthy human skin and in lesional and nonlesional skin of patients with skin diseases. Currently, the most common approach is to obtain skin biopsies and to visualize $\mathrm{M\Phi}$s by immunohistochemistry. Skin biopsies, however, come with several important limitations, which include scarring, the risk of infection and bleeding, and artificial findings caused by the use of local anesthesia. In addition, histopathological analyses of skin biopsies are not well suited for characterizing $\mathrm{M\Phi}$ functions such as phagocytosis and for long-term monitoring of $\mathrm{M\Phi}$ distribution in the skin.

Fluorescence lifetime imaging (FLIM) employs NAD(P)H and fluorescence decay parameters of cellular compartments as specific indicators of cell types and phenotypes (*Alfonso-García et al., 2016*; *Heaster et al., 2021*). Combined with two-photon tomography, two-photon excited fluorescence lifetime imaging (TPE-FLIM) allows for label-free and noninvasive imaging of dermal cells. For instance, TPE-FLIM allows for in vitro imaging of mast cells, fibroblasts, neutrophils, and dendritic cells and in vivo imaging of mast cells in human skin (*Kröger et al., 2020*). Whether or not TPE-FLIM can be used to visualize human skin $\mathrm{M\Phi}$s, their M1 and M2 phenotypes, and their functions, is currently unknown. There are, however, several independent lines of evidence that support this approach: First, previous studies have shown that TPE-FLIM can distinguish $\mathrm{M\Phi}$s from other dermal cells and ECM, without prior labeling (*Kröger et al., 2020*). Second, the capillaries of the papillary dermis, which often are in close proximity to $\mathrm{M\Phi}$s, show distinct TPE-FLIM signatures and are readily visualized (*Shirshin et al., 2017*). Third, M1 and M2 $\mathrm{M\Phi}$s come with unique cytokine patterns, and the TPE-FLIM signatures of these cytokines and patterns could help to tell the two phenotypes apart. Finally, TPE-FLIM can distinguish between functional states of dermal cells, for example, resting and activated mast cells in vivo, $\mathrm{M\Phi}$s ex vivo (*Kröger et al., 2020*), and T-cell activation in vitro (*Walsh et al., 2021*) may, therefore, potentially allow for monitoring $\mathrm{M\Phi}$ functions in vivo (*Szulczewski et al., 2016*). Taken together, the morphological features of skin $\mathrm{M\Phi}$s, their localization in the skin, and the expected differences in fluorescence decay parameters between $\mathrm{M\Phi}$ phenotypes as well as between other dermal cells, make TPE-FLIM a promising strategy for their detection (*Yakimov et al., 2019*).

Here, we first investigated human skin $\mathrm{M\Phi}$s, in vitro with clear M1 and M2 phenotypes, for their TPE-FLIM properties and how these differ from those of the main components of the ECM and other dermal cells such as fibroblasts, mast cells, and dendritic cells. We then applied the identified $\mathrm{M\Phi}$ TPE-FLIM signatures to investigate M1 and M2 $\mathrm{M\Phi}$s and their phenotypes in human skin biopsies, combined with traditional immunohistochemistry-based visualization. Finally, we used TPE-FLIM in vivo in humans to study skin $\mathrm{M\Phi}$s, their phenotypes, and functions, and we developed, tested, and characterized TPE-FLIM signature-based machine learning algorithms for the detection of skin $\mathrm{M\Phi}$s.

## Results

### In vitro monocyte-derived M1 and M2 $\mathrm{M\Phi}$s show distinct TPE-FLIM parameters

The TPE-FLIM images of monocytes isolated from human peripheral blood mononuclear cells (PBMCs) showed a round morphology (diameter of up to 10 µm) with a barely visible nucleus, homogeneously

distributed cell content, and regular borders with no membrane extensions (*Figure 2—figure supplement 1*). $M\Phi$s differentiated from PBMC and polarised toward M1 $M\Phi$s with interferon-γ (IFN-γ; *n*=21) and toward M2 $M\Phi$s with interleukin-4 (IL-4; *n*=27) were similar in size, ranging 10–12 µm (*Figure 2a–c*). M1 and M2 $M\Phi$s showed comparable overall TPE-AF intensities, but they differed significantly in several other features. M1 $M\Phi$s also showed numerous bright spots (typical size is 2–3 µm), likely vacuoles and mitochondria, had less visible borders, and exhibited higher TPE-AF intensity than M2 $M\Phi$s. In contrast, M2 $M\Phi$s were characterized by distinct borders with filopodia (*Figure 2b*; *Figure 2—figure supplement 2*), which were rarely seen in M1 $M\Phi$s (*Figure 2a*).

M1 and M2 $M\Phi$s also differed in their TPE-FLIM parameters $\tau_1$, $\tau_2$, and $\tau_m$ (*Figure 2d*). TPE-AF decay times were significantly shorter in M1 $M\Phi$s (*n*=21) than in M2 $M\Phi$s (*n*=27; *p*<0.05), and both $M\Phi$s differed significantly, in their TPE-FLIM parameters, from monocytes (*n*=15; *p*<0.05; *Table 1*).

IgG stimulation of IgG immune complex-sensitized M1 and M2 $M\Phi$s resulted in the release of inflammatory mediators, but did not lead to significant changes or reveal additional differences in TPE-FLIM parameters 2 and 5 days after differentiation of PBMC into $M\Phi$s (data not shown). Taken together, these findings indicate that monocyte-derived M1 $M\Phi$s and M2 $M\Phi$s can be identified and distinguished in vitro by their distinct TPE-FLIM signatures.

## $M\Phi$s isolated from periocular skin show TPE-FLIM parameters that are similar to those of in vitro monocyte-derived M1 $M\Phi$s or M2 $M\Phi$s

Human $M\Phi$s isolated from periocular skin and analyzed by immunohistochemistry were irregularly shaped, with poorly defined borders, 8–10 µm in size, pericentral nuclei of 5–6 µm diameter with low fluorescence intensity, heterogeneously and irregularly distributed cellular content, and they exhibited a bright fluorescence multivacuolated cytoplasm with ≈1 µm diameter small bright spots, presumably related to mitochondria and/or vacuoles (*Figure 2c*). Based on their TPE-FLIM parameters, dermal $M\Phi$s fell into two significantly different groups (*Figure 2d*): group 1 (*n*=34), with stronger TPE-AF intensity (≈3000±500 photons/mW) and shorter lifetimes, and group 2 (*n*=28), with a weaker TPE-AF intensity (≈800±200 photons/mW) with longer lifetimes (*Figure 2c*; *Figure 2—figure supplement 3*). The profiles of dermal $M\Phi$s in groups 1 and 2 were similar to those of monocyte-derived M1 $M\Phi$s and M2 $M\Phi$s, respectively (*Figure 2d*; *Table 1*). The biggest differences between group 1/M1 $M\Phi$s and group 2/M2 $M\Phi$s were shorter $\tau_1$ and $\tau_m$ as well as larger size (10.9±0.6 µm) in the former as compared to the latter (9.8±1.2 µm; *p*<0.05; *Figure 2c and d*; *Table 1*). This suggests that human skin $M\Phi$s, based on their in vitro TPE-FLIM signatures, can be assigned to one of two phenotypes, where the first is similar to that of monocyte-derived M1 $M\Phi$s and the second is similar to that of monocyte-derived M2 $M\Phi$s.

It should be noted that the TPE-FLIM parameters were stable over the measurements that took up to 1 hr for in vitro M1 and M2 $M\Phi$s isolated from periocular skin (*Figure 2—figure supplement 4*). Their TPE-FLIM values vary within the standard deviation shown in *Table 1*.

## TPE-FLIM can distinguish between $M\Phi$s and other cells

To prove that the recorded TPE-FLIM signatures are unique for M1- and M2-polarized $M\Phi$s (*Figure 2*), we performed TPE-FLIM measurements of other dermal cells in vitro, such as mast cells, dendritic cells, fibroblasts, monocytes, and neutrophils. Their TPE-FLIM parameters, summarized in *Table 1*, are markedly different from those of the established signatures of M1- and M2-polarized $M\Phi$s. Thus, in addition to size, morphology, and internal vacuole structure, M1- and M2-polarized $M\Phi$s can be distinguished, from each other and other cells, by distinct TPE-FLIM parameters, a prerequisite for the visualization of skin $M\Phi$s ex vivo and in vivo. *Table 1* is an extension of the results shown in *Kröger et al., 2020*.

## Immunohistochemistry confirms TPE-FLIM detection of M1 and M2 $M\Phi$s in human skin ex vivo

To test if the TPE-FLIM signatures established in vitro identify M1- and M2-polarized $M\Phi$s in human skin, we sequentially analyzed dermal biopsies by TPE-FLIM and conventional immunohistochemistry. The application of in vitro $M\Phi$ signatures to TPE-FLIM analyses of 13 human skin biopsy cryo-sections identified two distinct cell populations: The first showed a short mean fluorescence lifetime $\tau_m$ and high TPE-AF intensity, a feature of M1 $M\Phi$s (*Table 1*, *Figure 2d*, *Figure 3a*); the second population

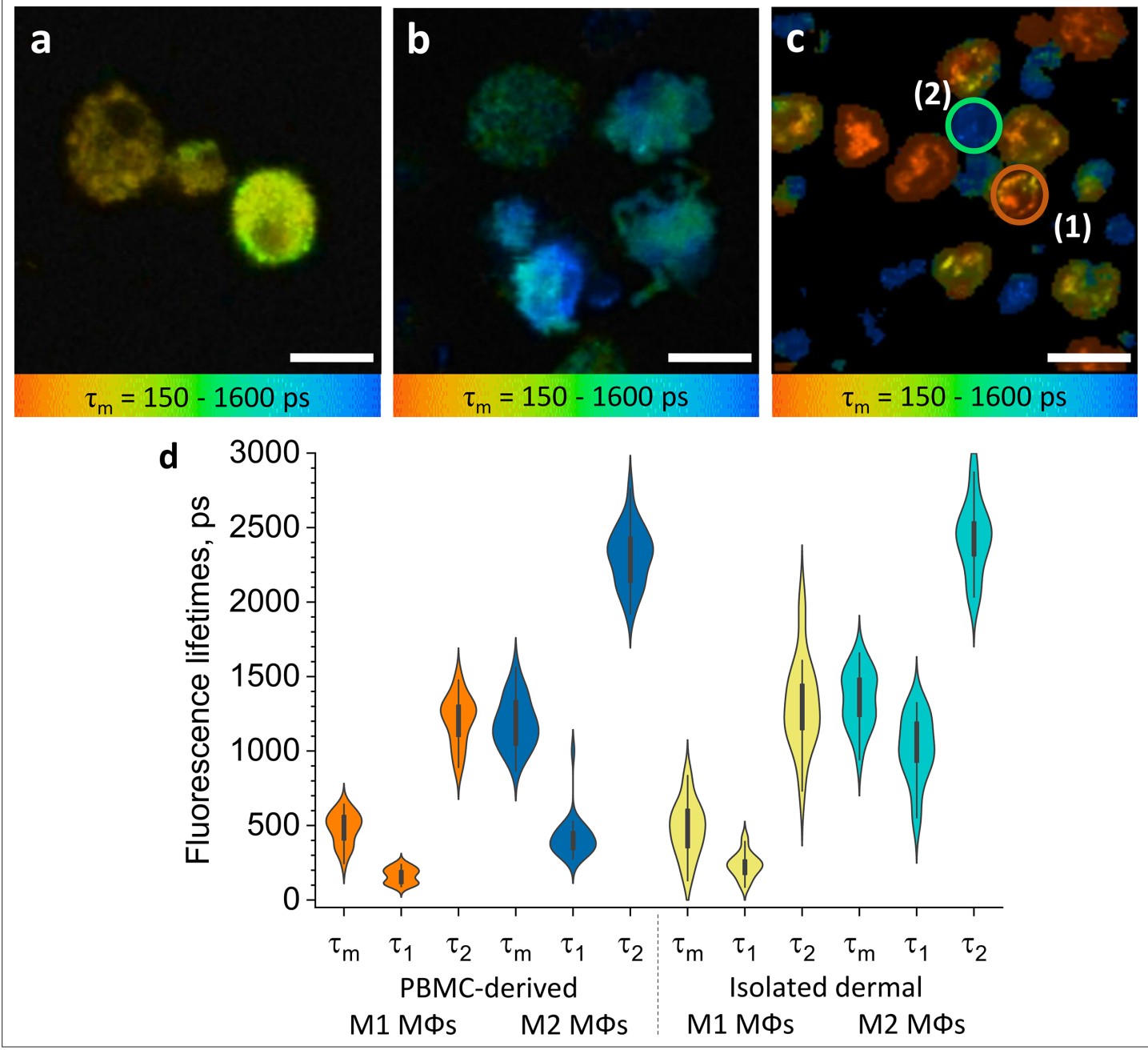

**Figure 2.** MΦs polarised from PBMC and isolated dermal MΦs show distinct TPE-FLIM signatures. TPE-FLIM $\tau_m$ images (mean fluorescence lifetime $\tau_m$ in the 150–1600 ps range) of monocyte-derived M1-polarised (IFN-γ) MΦs (**a**), monocyte-derived M2-polarised (IL-4) MΦs (**b**), and isolated human dermal M1 MΦs (1) and M2 MΦs (2) (**c**). Scale bar: 10 μm. The distribution of TPE-FLIM parameters $\tau_1$, $\tau_2$, and $\tau_m$ for monocyte-derived M1-polarised MΦs (*n*=21, orange), M2-polarised MΦs (*n*=27, dark blue), and isolated dermal M1 MΦs (*n*=34, yellow), M2 MΦs (*n*=28, light blue) (**d**). The boxplot represents 25–75% of the values. PBMC, peripheral blood mononuclear cell; TPE-FLIM, two-photon excited fluorescence lifetime imaging.

The online version of this article includes the following figure supplement(s) for figure 2:

**Figure supplement 1.** TPE-FLIM visualization of PBMC and histogram of TPE-FLIM parameter.

**Figure supplement 2.** In vitro 2D segmentation for MΦs from PBMC.

**Figure supplement 3.** In vitro segmentation for MΦs from dermal tissue.

**Figure supplement 4.** Time stability of TPE-FLIM parameters for MΦs isolated from periocular skin in vitro.

**Table 1.** TPE-FLIM parameters for investigated dermal and epidermal cells.

TPE-FLIM parameters $\tau_1$, $\tau_2$, $\tau_m$, $a_1/a_2$ and TPE-AF intensity of monocyte-derived M1 and M2 MΦs; dermal M1 and M2 MΦs isolated from the skin measured in vitro; M1 (CD68) and M2 (CD163) MΦs measured ex vivo in human skin cryo-sections; M1 and M2 MΦs observed on the forearm of healthy volunteers in vivo; monocytes; resting and activated human skin mast cells; dendritic cells; fibroblasts and neutrophils in vitro.

| | | Number of cells | $\tau_m$ in ps | $\tau_1$ in ps | $\tau_2$ in ps | $a_1/a_2$ | TPE-AF intensity, photons / mW |
|---|---|---|---|---|---|---|---|
| in vitro | Monocyte-derived M1-polarised MΦs | 21 | 479±106 | 163±50 | 1,209±161 | 2.4±0.6 | 600±100 |
| in vitro | Monocyte-derived M2-polarised MΦs | 27 | 1,185±170 | 417±134 | 2,305±194 | 2.3±0.5 | 500±100 |
| in vitro | M1 isolated dermal MΦs | 34 | 461±175 | 225±84 | 1,289±278 | 4.8±3.4 | 3,000±500 |
| in vitro | M2 isolated dermal MΦs | 28 | 1,281±155 | 807±250 | 2,352±229 | 2.2±1.1 | 800±200 |
| ex vivo | M1 MΦs (CD68) | 8 | 458±50 | 190±38 | 1,504±133 | 4.1±0.7 | 3,000±500 |
| ex vivo | M2 MΦs (CD163) | 12 | 1,369±201 | 498±129 | 2,267±155 | 1.1±0.4 | 700±300 |
| in vivo | M1 MΦs | 35 | 477±105 | 196±40 | 1,698±172 | 5.0±2.8 | 686±165 |
| in vivo | Phagocytosing M1 MΦs | 2 | 195±44 | 105±10 | 1,272±89 | 14.7±4.5 | 1,100±150 |
| in vivo | M2 MΦs | 25 | 1,407±60 | 442±54 | 2,458±90 | 1.2±0.2 | 360±155 |
| in vitro | PBMC-derived monocytes | 15 | 989±111 | 491±130 | 2,025±301 | 1.8±0.5 | 700±130 |
| in vitro | Resting mast cells | 43 | 1,248±287 | 533±266 | 2,289±317 | 1.5±0.5 | 1,300±400 |
| in vitro | Activated mast cells | 13 | 862±268 | 288±130 | 1,920±287 | 2.5±2.0 | 900±200 |
| in vitro | Dendritic cells | 14 | 1,265±180 | 434±188 | 2,578±328 | 1.6±0.2 | 538±258 |
| in vitro | Fibroblasts | 6 | 921±81 | 429±51 | 1,983±137 | 0.5±0.1 | 469±137 |
| in vitro | Neutrophils | 21 | 1,074±109 | 714±250 | 1,795±600 | 1.5±0.5 | 500±115 |

showed longer $\tau_m$ and significantly lower TPE-AF intensity, typical for M2 MΦs (**Table 1**, **Figure 2d**, **Figure 3c**). CD68-staining for M1 MΦs and CD163-staining for M2 MΦs confirmed that short $\tau_m$ cells with high TPE-AF intensity were, indeed, M1 MΦs (**Figure 3b**) and that cells with longer $\tau_1$, $\tau_2$, and $\tau_m$ with low TPE-AF intensity were, indeed, M2 MΦs (**Figure 3d**, **Figure 3—figure supplement 1**, **Table 1**).

CD68-positive dermal M1 MΦs showed a heterogeneous appearance, ranging from flat and spindle-shaped vessel lining to big intravascular with irregular borders and an irregular nucleus (**Figure 3b**). The TPE-FLIM image of CD163-positive M2 MΦs show round to elliptically shaped cells with a significantly lower TPE-AF intensity (**Figure 3d**). Of nine cells with a TPE-FLIM M1 MΦ signature, eight cells stained positive for CD68, and all CD68-positive cells had a TPE-FLIM M1 MΦ signature. As for M2 MΦs, all cells with a TPE-FLIM M2 MΦ signature (12 of 14) were CD163-positive, and all CD163-positive cells had a TPE-FLIM M2 MΦ signature.

## TPE-FLIM visualizes human skin M1 and M2 MΦs in vivo

Next, we used TPE-FLIM to assess the skin of 25 healthy individuals in vivo, and we identified and further characterized 35 and 25 MΦs with an M1 and M2 TPE-FLIM signature, respectively. In vivo, similar to biopsy sections, M1 and M2 MΦs were located in the papillary and reticular dermis at >80 μm depth (**Figure 4**) and showed a density of >100 MΦs/mm² (**Figure 1c and d**). M1 MΦs fell into three distinct groups and were either flat and spindle-shaped (**Figure 4a**), slightly dendritic (**Figure 4b**), or large and intervascular (**Figure 4c**). M2 MΦs, in human skin in vivo, were round and moderately dendritic (**Figure 4d**), and they had a higher TPE-AF intensity in vivo compared to the ECM, as previously reported in vitro (**Malissen et al., 2014**; **Njoroge et al., 2001**).

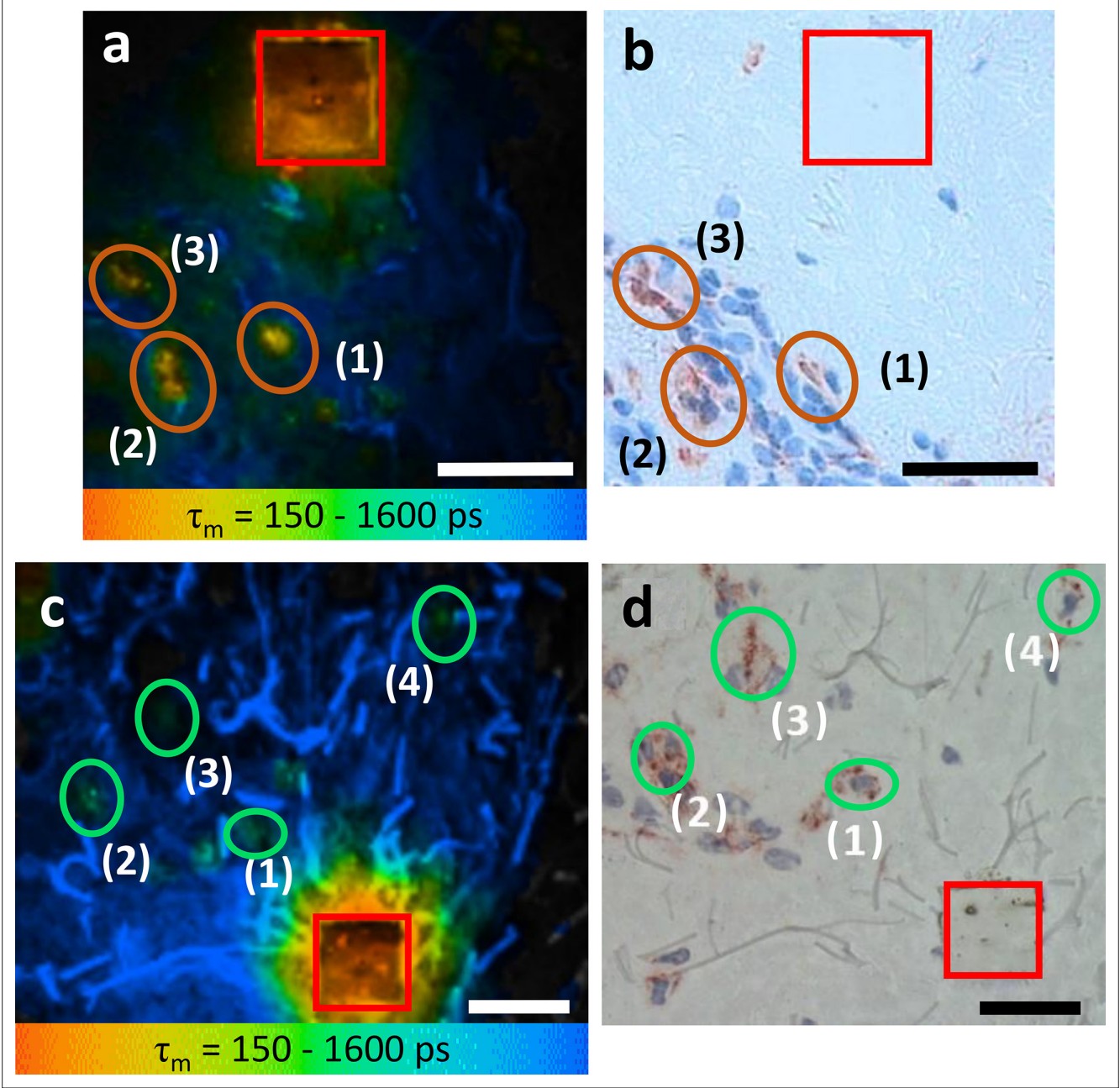

**Figure 3.** M1 and M2 MΦs ex vivo verified using TPE-FLIM parameters and immunohistochemistry-based bright field microscopy. Side by side comparison of TPE-FLIM $\tau_m$ images (mean fluorescence lifetime $\tau_m$ in the 150–1600 ps range), which were measured label-free and then stained with CD68-antibody for M1 MΦs (**a**), and CD163-antibody for M2 MΦs (**c**) and corresponding bright field microscopic images (**b**) and (**d**). The excitation wavelength is 760 nm and laser power is 4 mW (**a**) and 2 mW (**c**). The M1 and M2 MΦs are marked with ellipses in (**a, b**) and in (**c, d**), respectively. The laser-burned labels (28×28 µm²) are marked in red. The suspected (**a, c**) and staining-proved (**b, d**) MΦs are marked with number (1, 2, 3, and 4). More M2 MΦs are observed in (**d**) compared to (**c**) due to the staining and visualization of the entire biopsy volume in (**d**) and limited imaging plane of the two-photon tomograph (1.2–2.0 µm) in (**c**). Images have been rotated and zoomed to match their orientation and size. Scale bar: 30 µm. TPE-FLIM, two-photon excited fluorescence lifetime imaging.

The online version of this article includes the following figure supplement(s) for figure 3:

**Figure supplement 1.** Ex vivo segmentation for MΦs from skin biopsies.

**Figure supplement 2.** Mast cell-specific staining with tryptase ex vivo – negative control.

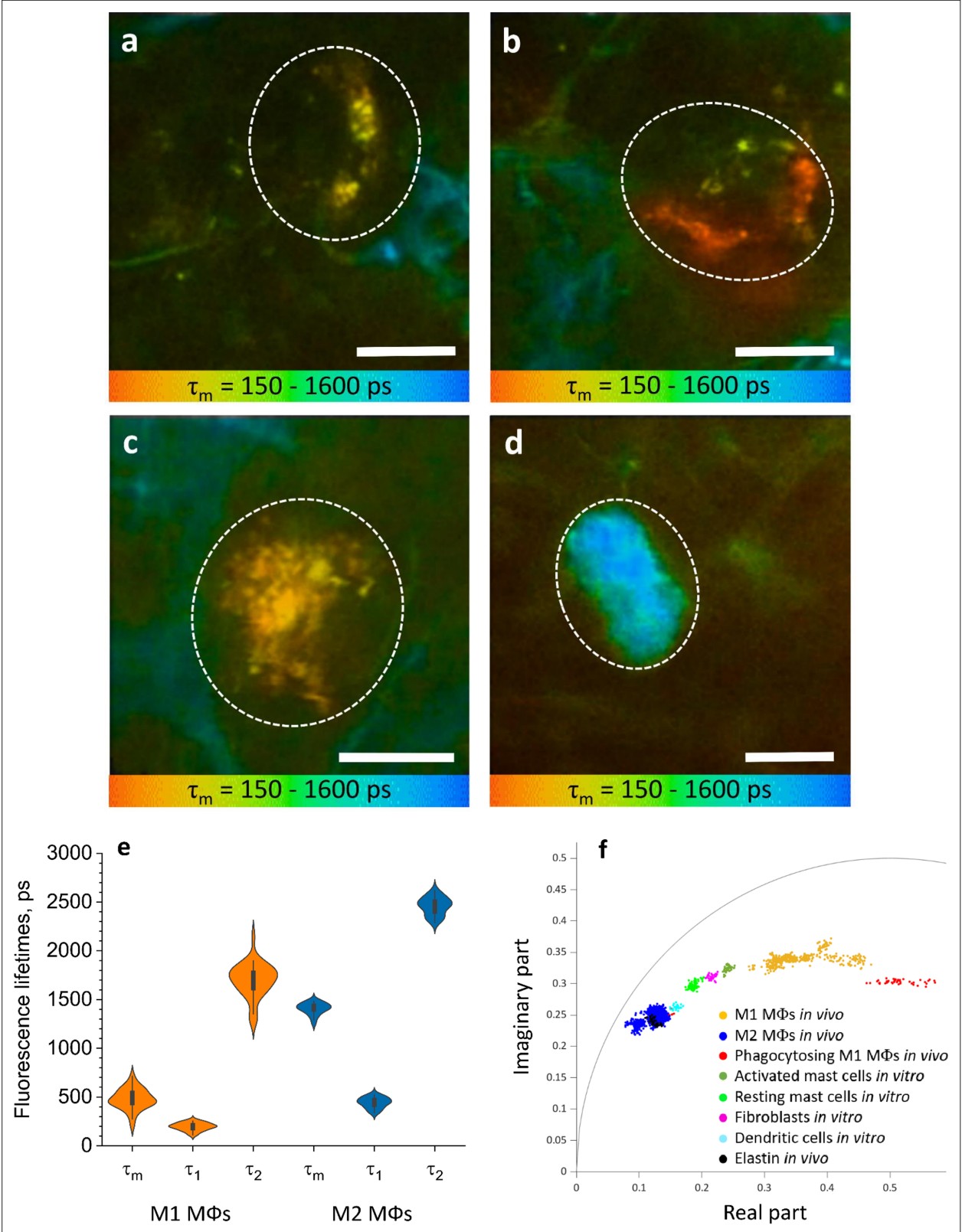

**Figure 4.** MΦs are visualized and categorised by TPE-FLIM signatures in vivo. TPE-FLIM in vivo images of potential perivascular flat spindle shaped M1 MΦ (**a**), of suspected slightly dendritic M1 MΦ in the depth 90 μm (**b**) large intervascular M1 MΦ with membrane extensions (**c**) and in vivo dermal cells resembling M2 MΦ were observed with a significantly longer mean fluorescence lifetime $\tau_m$ compared to M1 MΦs and less pronounced TPE-AF intensity (**d**), showing mean fluorescence lifetime $\tau_m$ in color gradient from 150 to 1600 ps. Scale bar: 10 μm. The histogram shows the distribution of

*Figure 4 continued*

TPE-FLIM parameters for M1 MΦs (*n*=35, orange) and M2 MΦs (*n*=25, blue) measured in vivo in human skin (**e**). The boxplot represents 25–75% of the values. The phasor plot has a threshold at 0.9 of the maximum intensity and shows a summary of 12 M1, 2 phagocytosing M1 MΦs and 12 M2 MΦs in vivo (**f**), where M1 MΦs are in orange and M2 MΦs in blue and phagocytosing M1 MΦs in red, the other dermal components are shown from in vitro measurements. The in vivo images (**a–d**) were recorded at 760 nm excitation wavelength, 50 mW laser power and 6.8 s acquisition time, in the depth of 80–100 μm on the volar forearm skin area of 25 healthy human subjects. TPE-AF, two-photon excited autofluorescence; TPE-FLIM, two-photon excited fluorescence lifetime imaging.

The online version of this article includes the following figure supplement(s) for figure 4:

**Figure supplement 1.** In vivo segmentation for MΦs in human skin.

**Figure supplement 2.** Time stability of TPE-FLIM parameter for M1 MΦ in vivo.

**Figure supplement 3.** Time stability of TPE-FLIM parameter for M2 MΦ in vivo.

**Figure supplement 4.** TPE-FLIM allows for visualization of potentially phagocytosing M1 MΦs in vivo.

**Figure supplement 5.** Decision tree model.

**Figure supplement 6.** ROC curves of decision tree models.

The TPE-FLIM parameters of in vivo M1 MΦs were in agreement with those of in vitro monocyte-derived and dermal M1 MΦs and ex vivo M1 MΦs (***Figure 4e***, ***Table 1***). M2 MΦs in vivo have longer $\tau_m$ fluorescence lifetimes compared to in vitro and ex vivo experiments. Yet, the $\tau_1$ and $\tau_2$ were in agreement with in vitro PBMC-derived monocytes, and the size and morphological parameters were in line with what is expected in M2 MΦs. The 2D segmentation in ***Figure 4—figure supplement 1*** shows the distinction of M1 and M2 MΦs presented in ***Figure 4a–d***, and the phasor plot in ***Figure 4f*** shows that M1 and M2 MΦs could be distinguished from each other and from other dermal cells and ECM.

It should be noted that the TPE-FLIM parameters were stable over the measurements that took up to 30 min for in vivo M1 and M2 MΦs in the skin (***Figure 4—figure supplements 2 and 3***). The TPE-FLIM values vary within the standard deviation shown in ***Table 1***.

## TPE-FLIM can potentially distinguish resting from phagocytosing human skin M1 MΦs in vivo

Phagocytosing skin M1 MΦs are characterized by an increase in cell size (***May and Machesky, 2001***), enhanced vacuolization (***Cheng et al., 2019***), a shift of TPE-FLIM parameters toward shorter fluorescence lifetime values (***Yakimov et al., 2019***), acidification (***Li et al., 2017***; ***Teixeira et al., 2018***) and thus stimulated ROS production, different from those of resting M1 and M2 MΦs. A dermal cell matching all these criteria indicating phagocytosis is visualized in vivo using TPE-FLIM and presented in ***Figure 4—figure supplement 4***. This cell is located in the reticular dermis, has an enlarged size (≈25 μm) and an oval shape, similar to the resting M1 MΦ in ***Figure 4c***, pronounced vacuole structure and short TPE-FLIM lifetime indicative for phagocytosing M1 MΦ. Of 37 dermal M1 MΦs analyzed in vivo, 2 showed possible phagocytosis activity, and both were located in the reticular dermis below 100 μm of depth.

## Classification algorithm to identify MΦs in the skin

To separate M1 and M2 MΦs from other dermal cells, we developed a classification algorithm, which used the decision tree (***Figure 4—figure supplement 5***) and automatically classified MΦs based on their TPE-FLIM parameters and morphological features. The parameters of the decision tree were improved using hyperparameter optimization. The splitting method in the nodes of the decision tree classifier is chosen to be entropy impurity. To ensure the optimal quality of a split in the node of the decision tree, the following requirements had to be fulfilled: the minimal samples for a split are 2, the maximum depth of the tree is 9, and the samples had equal weight for the model classifying M1 and M2 MΦs. The independent TPE-FLIM parameters $\tau_1$, $\tau_2$, $a_1$, and $a_2$ and the dependent TPE-FLIM variables $\tau_m$, $\tau_2/\tau_1$, $a_1/a_2$, $(a_1-a_2)/(a_1+a_2)$ have been used for the best classification results, as can be seen in the decision tree model in ***Figure 4—figure supplement 5***. The ground truth was established by classification of in vitro and ex vivo MΦs with known phenotype resulting in 0.95±0.05 sensitivity and 0.97±0.06 specificity. When MΦs were classified as one group against other dermal cells, the sensitivity was 0.81±0.03 and the specificity was 0.81±0.03. Our algorithm also distinguished M1 MΦs from M2 MΦs and other cells, with a sensitivity of 0.88±0.04 and a specificity of 0.89±0.03. For

distinction of M2 $\mathrm{M\Phi}$s from M1 $\mathrm{M\Phi}$s and other cells, the sensitivity was 0.82±0.03 and the specificity was 0.90±0.03; receiver operating characteristic (ROC) is shown in *Figure 4—figure supplement 6*. Additionally, a fivefold cross-validation was additionally executed with these results: (0.87; 0.92; 0.87; 0.89; and 0.94), the mean of k-fold scores using cross_val_score method is 0.90 with a score of 1 describing evenly distributed data.

## Discussion

This is the first in vivo study to show that human skin $\mathrm{M\Phi}$s can be distinguished from other dermal cells and quantified through visualization with label-free, completely noninvasive TPE-FLIM. This risk-free approach also allows for the identification of $\mathrm{M\Phi}$ phenotypes, that is, M1 and M2 $\mathrm{M\Phi}$s, and for the characterization of their functional stage, that is, resting versus phagocytosing M1 $\mathrm{M\Phi}$s. Finally, TPE-FLIM can be used to implement sensitive and specific machine learning algorithms for $\mathrm{M\Phi}$ detection in the skin.

Our initial work with CD-14 positive monocytes isolated from PBMC and then differentiated and polarised toward M1 (IFN-γ) and M2 (IL-4) $\mathrm{M\Phi}$s was needed to establish their TPE-FLIM parameters. In fact, it showed that $\mathrm{M\Phi}$s are fluorescence-active, and, more importantly, that their TPE-FLIM parameters are among the best differentiators of M1 ($\tau_m$=479±106) and M2 ($\tau_m$=1,185±170) $\mathrm{M\Phi}$s. M1 $\mathrm{M\Phi}$s are associated with a slightly higher TPE-AF intensity (*Table 1*), which is a prominent indicator for the metabolic stress of the cell on account of a shift in lifetimes by changing amounts of free and bound NAD(P)H (*AlShabany et al., 2016*) and generation of ROS in mitochondria, phagosomal vacuoles, and the cell membrane (*Datta et al., 2015*). Additionally to NAD(P)H, autofluorescence of lipids and other cell compartments was recorded. TPE-AF intensity is a parameter with limitation due to the nonlinear imaging technique. There is no linear correlation between excitation and emission intensity, also it is reduced due to scattering and absorption in the skin. The metabolism of LPS-induced M1 $\mathrm{M\Phi}$s is characterized by higher glycolysis, indicating a shift toward shorter fluorescence lifetime (*Li et al., 2020*; *Orihuela et al., 2016*). The longer fluorescence lifetime $\tau_2$ in M2 $\mathrm{M\Phi}$s is best explained by oxidative phosphorylation and the emergence of fluorophores caused by fatty acid oxidation (*Viola et al., 2019*). NAD(P)H fluorescence is ubiquitously present in cells and exhibits the continuum of lifetimes in the 360–3400 ps range. Therefore, changes in TPE-FLIM parameters are likely a reason of the metabolic changes of the $\mathrm{M\Phi}$s. Free NAD(P)H has a short lifetime of 360 ps. For bound NAD(P)H, longer lifetimes up to 2–4 ns have been reported (*Alfonso-García et al., 2016*). A higher ratio of bound to free NAD(P)H is associated with M2 $\mathrm{M\Phi}$s resulting in longer TPE-FLIM parameters, while a lower ratio of bound to free NAD(P)H is associated with M1 $\mathrm{M\Phi}$s resulting in faster TPE-AF decay (*Blacker et al., 2014*). Thus, a strong indicator for the $\mathrm{M\Phi}$ polarization is the TPE-FLIM parameters of monocytes in between cohorts of $\mathrm{M\Phi}$s.

It was observed that the quantity of fluorescence lifetimes in $\mathrm{M\Phi}$s is vastly varying between M1 and M2 $\mathrm{M\Phi}$s. Regarding the $\mathrm{M\Phi}$ polarization, the paradigm shifts toward a less strict classification compared to M1 (IFN-γ/LPS-polarized) and M2 (IL-4-polarized). While this categorization is useful in clinical terms, the multitude of parameters leading to the differentiation process leaves $\mathrm{M\Phi}$s with wide-ranging properties both in expression of markers and also in appearance and TPE-FLIM parameters (*Murray, 2017*).

M1 (IFN-γ/LPS-polarized) $\mathrm{M\Phi}$s rely on the NADH oxidase and production of ROS, which is shown by fluorescent lifetimes of under 250 ps and mitochondrial fission, which can indicate the bright spots, whereas M2 (IL-4-polarized) rely on oxidative phosphorylation and fatty acid oxidation, together with mitochondrial fusion, it can explain the homogeneous appearance of M2 $\mathrm{M\Phi}$s (*Ramond et al., 2019*; *Swindle et al., 2002*; *Xu et al., 2016*).

Translation of our in vitro findings to $\mathrm{M\Phi}$s isolated from human skin confirmed that the latter share the TPE-FLIM signatures of the former, with shorter $\tau_m$ in dermal M1 $\mathrm{M\Phi}$s and longer $\tau_m$ in dermal M2 $\mathrm{M\Phi}$s. The classification into M1 and M2 $\mathrm{M\Phi}$s in vitro based on their distinct TPE-FLIM parameters was supported by their differences in size, morphology and internal vacuole structure. That the $\tau_1$ lifetime of dermal M2 $\mathrm{M\Phi}$s is longer than that of monocyte-derived M2 $\mathrm{M\Phi}$s is most likely due to the use of different polarization agents. $\mathrm{M\Phi}$ colony-stimulating factor (M-CSF) and IFNγ for M1 $\mathrm{M\Phi}$s and $\mathrm{M\Phi}$ colony-stimulating factor (M-CSF) and IL-4 for M2 $\mathrm{M\Phi}$s were used in PBMC-derived $\mathrm{M\Phi}$s and microenvironment effects, like inflammatory signals, UV exposure (*Kang et al., 1994*), and immune responses (*Theret et al., 2019*) influencing $\mathrm{M\Phi}$ functions, result in divergent fluorescence lifetimes

(*Zhang et al., 2014*; *Table 1*, *Figure 2d*). The most important outcome of our work with dermal MΦs was the establishment of their phenotype-specific TPE-FLIM signatures, a prerequisite for our subsequent in vivo studies and for comparing skin MΦs and other dermal cells.

In fact, the use of the TPE-FLIM signatures of M1 and M2 MΦs clearly allowed to distinguish them from mast cells, dendritic cells, fibroblasts, and neutrophils (*Table 1*, *Figure 4f*; *Kröger et al., 2020*). We controlled this by independent markers. For example, MΦs exhibited a higher fluorescence intensity compared to other dermal cells. In addition, they were larger than dermal mast cells, and their morphology was clearly different from that of neutrophils and fibroblasts. Vacuoles, a defining feature of MΦs, were linked to MΦ TPE-FLIM parameters, whereas granules, which identify mast cells, were not. In short, dermal MΦs, in their TPE-FLIM profiles, do not superimpose with other dermal cells or structures. The only exception, a partial overlap between M2 MΦs and elastin, is not relevant for the visualization of the former, as they are readily distinguished from the latter based on their morphology.

To confirm that ex vivo TPE-FLIM overlap with TPE-FLIM signatures of M1 and M2 MΦs in vitro, we sequentially analyzed cells in skin biopsy cryo sections with TPE-FLIM and conventional immunohistochemistry. Indeed, both approaches identified and distinguished matching MΦ populations, that is, M1 and M2 MΦs, with strong fluorescence intensity and spindle shape appearance of M1 MΦs and lower fluorescence intensity and longer fluorescence decay in M2 MΦs (*Figure 3*). Interestingly, M2 MΦs are often found in an area of higher density of unknown dermal cells, presumably fibroblasts, compared to M1 MΦs. It is suspected that M2 MΦs in conjunction with collagen-synthesizing fibroblasts are acting toward and aiding in dermal repair and regeneration. However, this approach also revealed some challenges that come with MΦ visualization by TPE-FLIM. For example, it was more difficult to visualize M2 MΦs than M1 MΦs in biopsies due to the high fluorescence intensity of elastin in dried tissue and other ECM components and a decreased signal-to-noise ratio. In *Figure 3d*, more CD163 positive M2 MΦs are visible compared to the corresponding TPE-FLIM image in *Figure 3c*, which is due to previously mentioned challenges and the limited imaging plane of the two-photon tomograph (1.2–2.0 µm) compared to a significantly thicker biopsy section (10 µm), which was stained in an entire depth and visualized by bright field microscopy. Importantly, immunohistochemistry confirmed our MΦ phenotype-specific TPE-FLIM signatures, and, in addition, confirmed that they distinguish MΦs from other dermal cells. Skin mast cells, for example, stained for tryptase, showed a distinct TPE-FLIM signature, confirming our recently reported findings on dermal mast cells in vivo (*Kröger et al., 2020*), and distinguished them from M1 and M2 MΦs (*Figure 3—figure supplement 2*).

When we turned to the visualization of skin MΦs in vivo, we had first to develop a search strategy. Important considerations included the preferred localization of MΦs in the papillary and reticular dermis, the orientation of the focal plane parallel to the skin surface, and the need for maximal cellular cross-section visualization, which requires a high-resolution adjustment in depth to reconstruct an entire cell structure. The application of this search algorithm successfully visualized M1 and M2 MΦs in human skin in vivo. The in vivo TPE-FLIM parameters of M1 MΦs were in agreement with those observed in vitro and ex vivo. M2 MΦs in vivo were characterized by longer mean fluorescence lifetime $\tau_m$ compared to in vitro and ex vivo (*Table 1*), which can be explained by the influence of environment (*Koo and Garg, 2019*; *Njoroge et al., 2001*). MΦs measured in vivo differ from cells measured in vitro by their simplified microenvironment (*Mosser and Edwards, 2008*) with missing growth factors and cytokines (*Melton et al., 2015*) and an elevated level of nutrients, which leads to different polarization of MΦs and different contributions of fluorescence lifetimes. Membrane extensions were harder to detect in vivo due to the obscuring effect of the surrounding ECM. We also observed that TPE-FLIM parameters in the same MΦs can vary depending on their cellular substructures, for example, the nucleus, vacuoles, cytoplasm, or membrane (*Figure 4a–d*). The phasor plot shows the relative position of the categories of MΦs and other cells. Furthermore, it shows the contributions of long and short fluorescence lifetimes, their discrepancies (*Table 1*) being due to the computational method and the harmonics at the repetition frequency of 80 MHz. Further investigations are needed to clarify how the location, morphology, and function of M1 and M2 MΦs influence their TPE-FLIM parameters and TPE-AF intensity in vivo. Such studies should also address the reasons for the differences in TPE-FLIM parameters between M1 and M2 MΦs, which may include differences in their metabolic pathways. M1 MΦs, for example, rely on NADH oxidase and production of ROS, which is linked to short fluorescence lifetimes below 250 ps and mitochondrial fission. M2 MΦs, on

the other hand, rely on oxidative phosphorylation and fatty acid oxidation, together with mitochondrial fusion (*Ramond et al., 2019*; *Swindle et al., 2002*; *Xu et al., 2016*).

The ability to visualize M1 and M2 MΦs by TPE-FLIM in vivo also makes it possible to explore how and why MΦs' morphology, location, and functions are linked. When activated, the cytoskeletal structure and cellular appearance of MΦs change, and this may also affect their TPE-FLIM parameters. M1 MΦs are elongated, with a dense actin network along the cortex. M2 MΦs are more spherical with more randomly distributed actin (*Porcheray et al., 2005*; *Vogel et al., 2014*; *Zhang et al., 2014*; *Figure 4b and d*). Actin reorganization in M1 and M2 MΦ polarization and activation lead to bigger filament bundles of the actin cytoskeleton, which reduces cell plasticity (*Colin-York et al., 2019*; *Pergola et al., 2017*). As reported by *Vogel et al., 2014*, MΦ migration in the skin depends on their polarization. M1 MΦs, due to changes in actin cytoskeleton, migrate less far than M2 MΦs. Our TPE-FLIM findings confirm this, as we detected M1 MΦs via their high fluorescence and short autofluorescence lifetimes primarily in close proximity to blood capillaries. The irregular appearance of M1 MΦ detected by TPE-FLIM is likely a consequence of polarization-specific changes of the cellular cytoskeleton (*McWhorter et al., 2013*). The only morphological feature observed in both, M1 and M2 populations of MΦs, is that they are moderately dendritic, possibly because such MΦs are in the process of polarization, prior to cytoskeletal changes (*Sica and Mantovani, 2012*), or because polarization in MΦs is reversible polarizing (*Sica and Mantovani, 2012*; *Yuan et al., 2017*). Future studies should characterize the influence of cytoskeletal changes on TPE-FLIM parameters in detail and use TPE-FLIM to assess the impact of age, gender, and disease on the ratio, localization, and function of M1 and M2 MΦs in the skin (*Fukui et al., 2018*).

Time stability measurements were performed in vitro on M1 and M2 MΦs isolated from periocular skin within 1 hr (*Figure 2—figure supplement 4*) and in vivo on M1 MΦ within 30 min (*Figure 4—figure supplements 2 and 3*).

TPE-FLIM parameters were stable and varied within the standard deviation range shown in *Table 1*. It was possible to visualize single cells over this long-term time period in vitro (*Figure 2—figure supplement 4*) and in vivo (*Figure 4—figure supplements 2 and 3*), yet no unprompted change in fluorescence lifetime was observed, given the laser power was not high enough to induce photoproducts or photobleaching of the fluorophores. The NAD(P)H-related TPE-FLIM parameter showed no change over the course of the measurement nor did the fluorescence intensity. The limitation of long-term in vitro measurements is that the cells cannot be heated and will eventually cool down to room temperature. The limitation of long-term in vivo measurements is inherent to the individual. It is technically possible to investigate one subject over the course of multiple hours, but it is almost impossible to find the same cell again in another measurement.

Another limitation of this study is the simplified separation between M1 and M2 MΦs without taking into consideration the mixed MΦ phenotype. As mentioned in the introduction, mixed phenotype of MΦs is primarily observed in diseases like melanoma, systemic sclerosis, Lupus, and in recovery from LPS tolerance. Thus, we do not expect a significant amount of mixed phenotype MΦs in healthy and asymptomatic skin. However, the MΦs misidentified by the decision tree (*Figure 2—figure supplement 3* and *Figure 4—figure supplement 1*) could potentially be MΦs of a mixed phenotype, which should be proved in additional experiment.

During phagocytosis, the generation of ROS by NAD(P)H oxidase leads to the highest degree of metabolic stress observed in M1 MΦs besides apoptosis (*Dupré-Crochet et al., 2013*; *Shirshin et al., 2019*), and ROS localization in vacuoles in phagocytosing M1 MΦs as a bactericidal mechanism (*Dupré-Crochet et al., 2013*; *Myers et al., 2003*). This is why phagocytosing M1 MΦs change their TPE-FLIM lifetimes toward shorter values and their vacuoles become visible as localized bright spots, which makes their in vivo detection possible (*Figure 4—figure supplement 4*, *Table 1*; *Cannon and Swanson, 1992*). The results shown here for possible phagocytosis are supported by the appearance (large size) and shortened fluorescence lifetime values for erythrophagocyting cells, presented by our group (*Yakimov et al., 2019*), resembling the cell shown in this study. The microenvironment in inflamed skin is known to be acidic with a pH<7.35 (*Haka et al., 2009*). Acidification is also known in phagocytosing MΦs (*Teixeira et al., 2018*). Together with the fact that fluorescence lifetime of fluorophores is shifted toward shorter values (*Li et al., 2017*) and cells especially MΦs produce ROS under phagocytosis and acidic conditions (*Slauch, 2011*), we have very compelling indications for the visualization of phagocytosing M1 MΦs. The influence of different phagocytosed materials in MΦs

should be investigated in the future. TPE-FLIM potentially allows for the detection of possible phagocytosing M1 $\mathrm{M\Phi}$s and is used as a confounder in the classification but cannot detect what is phagocytized. No internal structure was visible in TPE-FLIM images. The precision of measurements of cells and structures with short fluorescence lifetimes, such as phagocytosing $\mathrm{M\Phi}$s, could be improved by reducing the value of the instrument response function (IRF), which is <100 ps in our measurements.

The construction of the feature vector and the resulting hyperparameter optimized decision tree model (*Figure 4—figure supplement 5*) yielded proficient results for the automatised classification of M1 and M2 $\mathrm{M\Phi}$s, demonstrating that M1 and M2 $\mathrm{M\Phi}$s can be separated from each other and other cells in the skin with high accuracy, that is, sensitivity and specificity, without need of additional staining using a supervised machine learning approach. The decision tree model uses only the independent TPE-FLIM parameters $\tau_1$, $\tau_2$, $a_1$, and $a_2$ and the dependent TPE-FLIM variables $\tau_m$, $\tau_2/\tau_1$, $a_1/a_2$, $(a_1-a_2)/(a_1+a_2)$. This indicates that macrophages could be distinguishable completely from other cells without the use of morphologic parameters, thus reducing the degree of freedom and saving calculation and annotation time by software and physicians. The high accuracy for M1 macrophages is owed to the fact that M1 $\mathrm{M\Phi}$s have the shortest $\tau_1$ and $\tau_2$ fluorescence lifetimes, the highest ratio of $a_1/a_2$ and the highest fluorescence intensity of the cells in the model. M2 $\mathrm{M\Phi}$s can be misclassified in rare cases with resting mast cells and in vitro dendritic cells. Concluding from these results that the classification is only dependent on TPE-FLIM parameters and not on the morphology of dermal cells.

Ideally the data sets consist of the same amount of entries for every three classes (M1 $\mathrm{M\Phi}$s, M2 $\mathrm{M\Phi}$s, and other cells). In the experimental reality, those three classes are not evenly distributed and could lead to overemphasize of certain classes in the classifier model. It is shown here by the methods above that the data set is useable for the decision tree classifier. Overfitting of the relatively small data set is avoided with the parameters for the decision tree model, first by randomly splitting the data set into training and test set 10,000 times resulting in a small standard deviation and with the use of k-fold cross-validation resulting in a mean of the cross-validation score of 0.90, which equates to 90% accuracy. A cross-validation score of 1 describes perfectly even distributed data in all folds. The robustness and accuracy of this approach can be improved further, by the introduction of a depth-adjusted cell size and refined cell shape parameters and by increasing the number of in vivo $\mathrm{M\Phi}$s integrated into the algorithm and training data set.

## Materials and methods
### Two-photon excited fluorescence lifetime imaging

For imaging of human $\mathrm{M\Phi}$s, a two-photon tomograph (Dermainspect, JenLab GmbH, Jena, Germany), equipped with a tunable femtosecond Ti:sapphire laser (Mai Tai XF, Spectra Physics, USA, 710–920 nm, 100 fs pulses at a repetition rate of 80 MHz), was used at 3–5 mW for measurements of cells in vitro and skin biopsy sections ex vivo, as well as human dermis in vivo at 40–50 mW. The excitation wavelength was set to 760 nm, and a 410–680 nm band pass filter was used to detect two-photon excited autofluorescence (TPE-AF), whereas a 375–385 nm band pass filter was used to detect the second harmonic generation signals. The axial and lateral resolution was approximately 1.2–2.0 and 0.5 µm, respectively (*Breunig et al., 2013*). The screening depth covers the entire papillary dermis and part of reticular dermis (*Darvin et al., 2014*; *König, 2008*; *Kröger et al., 2020*).

Fluorescence decay of a specimen was recorded and analyzed in the SPCImage 8.0 software (Becker&Hickl, Berlin, Germany). TPE-FLIM data were fitted with a bi-exponential decay function. The TPE-AF intensity threshold was chosen depending on the signal-to-noise ratio, minimizing noise in the region of interest. The shift of the signal in relation to the instrument response function (IRF) was compensated. The typical IRF value was <100 ps. The TPE-AF decay curves were averaged over the central pixel of the region of interest and the 48 closest square neighbouring pixels (binning=3), resulting in a number of detected photons for each fluorescence decay curve larger than 5000. The TPE-AF decay parameters, decay lifetimes ($\tau_1$ and $\tau_2$) and amplitudes ($a_1$ and $a_2$), were used for the evaluation of the fluorescence lifetime distributions and 2D segmentation (*Shirshin et al., 2017*). The analyzed parameters were the mean lifetime, defined as $\tau_m=(a_1\tau_1+a_2\tau_2)/(a_1+a_2)$ and the ratios $\tau_2/\tau_1$, $a_1/a_2$ and $(a_1-a_2)/(a_1+a_2)$, which were used for 2D segmentation analysis. The TPE-FLIM data were also analyzed and represented as phasor plots, that are based on the transformation of the fluorescence decay data in the frequency domain, whereas the decay is described as amplitude and phase values of

the first Fourier component (*Digman et al., 2008*). The phasor plots' *x*-axis is described by the cosine of the phase value multiplied by the amplitude, the *y*-axis represents the sine of the phase value multiplied by the amplitude (*Lakner et al., 2017*; *Shirshin et al., 2019*). The position of the mean lifetime is on the secant from $\tau_1$ and $\tau_2$, the distance to the circle is given by the proportion of $a_1$ and $a_2$. The TPE-FLIM data were normalized to the maximum intensity and the threshold of 70% was set when analysing the phasor plots. The comparison of the bi-exponential fitting and phasor analysis in separation between cells subpopulations when treating the FLIM data was analyzed in *Shirshin et al., 2022*—it this work, we used both approaches to separate the M1 and M2 macrophages.

## FLIM data processing

The fluorescence decay curves were fitted with the bi-exponential decay model. Justification of the choice of the model and its comparison to the three exponential fitting is presented in the SI (Section FLIM data analysis). The absence of correlations between the fluorescence intensity and fluorescence decay parameters, as well as for fitting quality (assessed as $\chi^2$) and fluorescence decay parameters was additionally verified as described in the SI.

## Ethical considerations and study conduct

Volunteers for intravital imaging provided their written informed consent before participation. Skin samples taken from periocular skin surgery for $\mathrm{M}\Phi$ preparation and all human skin investigated in this study were used after written informed consent was obtained. Positive votes for the experiments have been obtained from the ethics committee of the Charité – Universitätsmedizin Berlin (EA1/078/18, EA4/193/18, and EA1/141/12), which were conducted according to the Declaration of Helsinki (59th WMA General Assembly, Seoul, October 2008).

## Study subjects

Twenty-five healthy volunteers (12 males and 13 females, 24–65 years old, skin type I–III according to *Fitzpatrick, 1988* classification) with asymptomatic volar forearm skin without preexisting health conditions were randomly selected for noninvasive in vivo measurements in the papillary dermis using TPE-FLIM. Visually impairing hair was removed with a scissor prior to measurements. The oil immersion objective of the microscope was connected to the skin via a 150 µm thick, 18 mm diameter cover glass (VWR, Darmstadt, Germany) with a ≈10 µl distilled water droplet between cover glass and skin. About 6–12 in vivo tomograms (different skin areas) were measured per subject, the investigated volume is from ≈70 µm depth in the papillary dermis to ≈130 µm depth in the reticular dermis with an image size of 150×150 µm². This adds up to (60×150×150) µm³ times 6–12 images, with a total volume of 0.008–0.016 mm³ of papillary and reticular dermis seen per subject, the average time spent was ≈30 min per subject and the acquisition time was 6.8 s per image. The volunteers were screened between October 2018 and November 2020.

## Investigation of human dermal $\mathrm{M}\Phi$s in vitro

Human dermal $\mathrm{M}\Phi$s were prepared from periocular tissue (*Botting et al., 2017*). Human periocular skin was digested in 2.4 U/ml dispase type II (Roche, Basel, Switzerland) at 4°C for 12 hr. The dermis was minced with scissors after removal of the epidermis and further digested in PBS containing $Ca^{2+}$ and $Mg^{2+}$ (Gibco, Carlsbad, CA) supplemented with 1% Pen/Strep, 5% FCS, 5 mM MgSO₄, 10 µg/ml DNaseI (Roche), 2.5 µg/ml amphotericin (Biochrom, Berlin, Germany,) 1.5 mg/ml collagenase type II (Worthington Biochemical Corp, Lakewood, NJ), and 0.75 mg/ml H-3506 hyaluronidase (Sigma-Aldrich, St. Louis, MO) at 37°C in a water bath with agitation for 60 min. The cell suspension was filtered using 300 and 40 µm stainless steel sieves (Retsch, Haan, Germany). Centrifugation at 300×*g* for 15 min at 4°C was applied next. The digestion cycle was repeated once. $\mathrm{M}\Phi$s were isolated by Pan Monocyte Isolation Kit (Miltenyi, Bergisch Gladbach, Germany) after washing in phosphate-buffered saline (PBS) w/o $Ca^{2+}$ and $Mg^{2+}$ (Gibco), and kept in basal Iscove's medium supplemented with 1% Pen/Strep, 10% FCS, 1% non-essential amino acids, 226 µM $\alpha$-monothioglycerol (all Gibco). For long-term cultures, after 24 hr recombinant human IL-4 (20 ng/ml) and hSCF (100 ng/ml) (both Peprotech, Rocky Hill, NJ) were added. Purity of $\mathrm{M}\Phi$ cultures was routinely checked to be >85% (*Nielsen et al., 2020*). For imaging, cells were used after 3 days in medium, washed two times with PBS before

seeding on 18 mm diameter microscope cover glass (VWR) for imaging in PBS containing $Ca^{2+}$ (Gibco) at room temperature.

## Investigation of peripheral blood monocytes in vitro

Peripheral blood monocytes were isolated from human blood using 15 ml Ficoll-Paque (VWR) centrifugation gradient. Centrifugation was performed at 1000×$g$ for 1 min, with added 9 ml heparin and filled to 50 ml with PBS. Centrifugation was then repeated at 1000×$g$ for 10 min, discarding the upper plasma layer and collecting the PBMC layer. The cells were washed two times with PBS and centrifuged at 350×$g$ for 10 min. The supernatant was discarded and cultured in 5 ml basal Iscove's medium supplemented with 1% Pen/Strep, 10% FCS (Biochtrom, Berlin, Germany) and subsequently incubated at 37°C and 5% $CO_2$ for 2 hr before seeded and imaged on an 18 mm diameter microscope cover glass (VWR) in PBS containing $Ca^{2+}$ (Gibco) at room temperature.

## Investigation of MΦs differentiated from peripheral blood monocytes in vitro

MΦs were differentiated from peripheral blood monocytes and polarised into M1 (IFNγ)-like state with MΦ colony-stimulating factor (M-CSF) and IFNγ and M2 (IL-4)-like state with MΦ colony-stimulating factor (M-CSF) and IL-4. For further stimulation, cells were incubated with LPS at 37°C for 24 hr prior to imaging. Due to a simplified environment with specific differentiation agents, the differentiation of monocytes was partially incomplete. Exemplary monocyte-derived MΦs appearing as M1 or M2 MΦs were measured and analyzed by TPT/FLIM. The requirement for M1 MΦs was a granular appearance and for M2 MΦs a dendritic appearance.

## Investigation of dendritic cells in vitro

CD14 positive PBMCs were used to differentiate dendritic cells by washing in PBS and centrifuging at 350×$g$ for 10 min two times. About 5 ml RPMI medium, supplemented with 1% Pen/Strep and 1% FCS (Biochtrom), was added. Tryptan Blue (Sigma-Aldrich) was used for counting the cells in a hemocytometer, seeded at 2.0×$10^6$ cells/ml and incubated for 2 hr at 37°C under 5% $CO_2$. Non-attached cells and the supernatant were discarded. Adding 500 µl basal Iscove's medium to the cells supplemented with 1% Pen/Strep, 1% glutamine, 5% HSA (all Gibco), 100 ng/ml IL-4, 100 ng/ml GM-CSF (both Peprotech) with medium change every second day for 6 days at 37°C. For TPE-FLIM imaging, the cells were seeded on 18 mm diameter microscope cover glass (VWR) in PBS containing $Ca^{2+}$ at room temperature.

## Preparation and cryo-sectioning of human skin for combined TPE-FLIM and histomorphometric analysis

Thirteen human skin biopsy cryo-sections were prepared and measured using the TPE-FLIM method to acquire TPE-FLIM parameters of suspected M1 and M2 MΦs. The skin biopsies were obtained from abdominal reduction surgery of four female patients (31, 33, 40, and 44 y. o., skin type II according to Fitzpatrick classification; *Fitzpatrick, 1988*). Punch biopsies of 6 mm diameter were obtained, frozen, and stored at –80°C before cryo-sectioning. Vertical histological cryo-sections of 10 µm thickness were prepared on a cryostat (Microm Cryo-Star HM 560, MICROM International GmbH, Walldorf, Germany) after embedding in a cryo-medium (Tissue Freezing Medium, Leica Biosystems Richmond Inc, Richmond, IL) and placed on 18 mm diameter microscope cover glasses (VWR). The anatomical condition of the biopsies was continuously examined using a transmission microscope (Olympus IX 50, Olympus K.K., Shinjuku, Tokyo, Japan).

Using TPE-FLIM, cryo-sections were searched for cells with MΦ-specific TPE-FLIM parameters and the corresponding TPE-FLIM images of suspected MΦs were recorded. To prove the measured cells are MΦs, the skin biopsies were labeled by irradiating a squared area of 28×28 µm$^2$ located near the suspected MΦs with a Ti:sapphire laser (Mai Tai XF, Spectra Physics, USA, 100 fs pulses at a repetition rate of 80 MHz) at a maximal power of 50 mW at 760 nm for 3 s. All incubations were performed at room temperature unless otherwise stated. In brief, sections were fixed for 10 min in cold acetone (–20°C) and rinsed in TBS (Agilent Technologies, Santa Clara, CA). For staining of MΦs, the MΦ-specific anti-CD68 (clone ab955) (Abcam, Cambridge, UK), Recombinant Anti-CD163 antibody [EPR14643-36] (clone ab189915) (Abcam) were used to account for M1

and M2 $\mathrm{M}\Phi$ phenotypes, respectively. Slides were rinsed three times with TBS, and endogenous peroxidase was blocked with 3% $H_2O_2$ in TBS for 5 min followed by incubation with anti-mouse EnVision+ labeled polymer (Agilent Technologies) for 30 min. Slides were rinsed in TBS as before and incubated with AEC substrate-chromogen (Agilent Technologies) for 10 min. Nuclei were counterstained with Mayer's hemalum solution (Merck, Darmstadt, Germany). Stained $\mathrm{M}\Phi$s have a brown-red color, which enables to visually distinguish them from other cells and the ECM. After the staining procedure, target $\mathrm{M}\Phi$s and squared labels of the skin sections were identified by light microscopy and overlaid with TPE-FLIM images matching an appropriate magnification and image orientations.

Specifically, CD68-stained M1 $\mathrm{M}\Phi$s were counted in the papillary dermis region in each biopsy, and an average of 209±25 cells/mm² for the papillary dermis and an average of 140±76 cells/mm² for the reticular dermis for a 10 µm deep cryo-section was observed (*Figure 1c*). The density of the CD163 stained M2 $\mathrm{M}\Phi$s was an average of 242±126 cells/mm² for the papillary dermis and an average of 107±60 cells/mm² for the reticular dermis for a 10 µm deep cryo-section (*Figure 1d*).

The $\mathrm{M}\Phi$s search algorithm we then used was similar to that recently presented by our group for the identification of resting and activated mast cells in the papillary dermis (*Kröger et al., 2020*) and included the following steps: first, the papillary dermis (≈60–100 µm depth for volar forearm) was explored for fluorescent spots of 10–15 µm in size with irregular shape and a membrane extension having bright spots of about 1–3 µm. The TPE-FLIM parameters of the suspected bright areas were measured and matched those of M1 and M2 $\mathrm{M}\Phi$s obtained in vitro and ex vivo.

To prove that the TPE-FLIM parameters of other dermal cells, which have detectable TPE-AF intensity, namely, mast cells and dendritic cells do not match or superimpose with TPE-FLIM parameters of $\mathrm{M}\Phi$s, negative control measurements were performed. The procedure was similar as described for the verification of $\mathrm{M}\Phi$s in skin biopsies using specific immunofluorescence, but six human skin cryosections were stained for the presence of mast cells and two for dendritic cells.

Staining of mast cells was done by blocking with serum-free protein followed by incubation for 1 hr with anti-tryptase antibody (clone AA1) diluted 1:1000 in antibody diluent (all Agilent Technologies). For staining of dendritic cells, anti-CD11c antibody (clone B-Ly6) (BD Biosciences, Franklin Lakes, NJ) was used after fixing the cryo-section for 10 min in cold acetone (–20°C) and rinsing with TBS.

## Statistical analysis and classification algorithm

Matlab R2016a (MathWorks, Natick, MA) was applied for descriptive statistics of all TPE-FLIM data. All results are indicated as mean ± standard deviation. Differences between distributions were compared using the nonparametric Kolmogorov-Smirnov test with a significance level of $\alpha$=0.05. The decision tree classifier was modelled using Scikit-learn 0.22 in a Python 3.7 environment (Python Software Foundation, Wilmington, DE). A randomised training set, consisting of 50% of the complete data set, was used for training and validating the test set 10,000 times. The true positive and true negative rates were calculated from the confusion matrix and describe the quality of the classification and indicate type I and type II errors. For the decision tree (*Breiman et al., 1984*), the TPE-FLIM parameters $\tau_1$, $\tau_2$, $\tau_m$, $\tau_1/\tau_2$, $a_1$, $a_2$, $a_1/a_2$, $(a_1-a_2)/(a_1+a_2)$, TPE-AF intensity, cell shape, and decay curve were used for each cell measured in vitro, ex vivo, and in vivo and hyperparametrically optimized (*Yang and Shami, 2020*). The feature vector was constructed as follows: 8 (TPE-FLIM parameters obtained after bi-exponential approximation of the decay curve), in total, 8 values. The $\mathrm{M}\Phi$ size was not included in the feature vector for the classification model, as $\mathrm{M}\Phi$s in vivo could have slightly different dimensions from those measured in vitro (in cell cultures) and ex vivo (in biopsies), caused by obscuring effects of surrounding dermal tissue. Here, 1 represents circular and 0 noncircular shape. The lifetimes calculated from the bi-exponential decay model were averaged over the whole cell, and the fluorescence intensity was normalized by optical power and averaged the pixel of interest and the 48 neighbouring square pixel.

In total, 110 $\mathrm{M}\Phi$s in vitro, 20 $\mathrm{M}\Phi$s ex vivo, 70 $\mathrm{M}\Phi$s in vivo (for M1/M2 ratio see *Table 1*), 59 mast cells in vitro, 17 mast cells ex vivo, 82 mast cells in vivo, 14 dendritic cells in vitro, 6 fibroblasts in vitro, and 21 neutrophils in vitro were used as input for the model (399 cells in total). Given data vectors from $x_i \in R^n$, i=1,…, $l$ and a label vector $y_i \in R^l$, where a decision tree recursively separates the data into two classes with the mode $m$ represented as $Q$. For each node a split $\theta = (j, t_m)$ decided with the feature $j$ and the threshold $t_m$. The node split the data into subsets $Q_{left}(\theta)$ and $Q_{right}(\theta)$.

$$Q_{left}(\theta) = (x, y) \, | x_j <= t_m$$

$$Q_{right}(\theta) = Q \backslash Q_{right}(\theta)$$

The impurity was calculated by the impurity function $H()$ at the mode $m$

$$G(Q, \theta) = \frac{n_{left}}{N_m} H\left(Q_{left}(\theta)\right) + \frac{n_{right}}{N_m} H\left(Q_{right}(\theta)\right)$$

With the parameters for minimized impurities, the subsets were recourse until $N_m=1$.

The return values of the classification were 0 for M1 $\mathrm{M\Phi}$s, 1 for M2 $\mathrm{M\Phi}$s, and 2 for other dermal cells, 0 for $\mathrm{M\Phi}$s and 1 for other dermal cells for node $m$ in the region $R_m$ and $N_m$ observation, the proportion of class $k$ observations in node $m$ is $p_{mk} = 1/N_m \sum\limits_{x_i \in R_m} I\left(y_i = k\right)$.

ROC curves served as a tool to determine the diagnostic abilities of the method, where the true positive rate was plotted against the false positive rate of the respective outcomes for both the categorization of $\mathrm{M\Phi}$s against other dermal cells and M1 $\mathrm{M\Phi}$s and M2 $\mathrm{M\Phi}$s against other dermal cells.

## Acknowledgements

The authors thank Evelin Hagen, Niklas Mahnke, and Loris Busch from Charité – Universitätsmedizin Berlin for their excellent technical support. The authors thank David Satzinger for the schematic illustration of $\mathrm{M\Phi}$s. Foundation for Skin Physiology of the Donor Association for German Science and Humanities (Marius Kröger, Johannes Schleusener, Martina C Meinke, Jürgen Lademann, Maxim E Darvin) Russian Science Foundation No. 19-75-10077 "Photonic and Quantum technologies. Digital medicine" (Evgeny A Shirshin).

## Additional information

### Funding

| Funder | Grant reference number | Author |
|---|---|---|
| Foundation for Skin Physiology | | Marius Kröger |
| Russian Science Foundation | 19-75-10077 | Evgeny A Shirshin |

The funders had no role in study design, data collection and interpretation, or the decision to submit the work for publication.

### Author contributions

Marius Kröger, Data curation, Software, Formal analysis, Investigation, Writing – original draft; Jörg Scheffel, Resources, Data curation, Investigation, Methodology, Writing - review and editing; Evgeny A Shirshin, Formal analysis, Validation, Methodology, Writing - review and editing; Johannes Schleusener, Formal analysis, Validation, Investigation, Methodology, Writing - review and editing; Martina C Meinke, Resources, Supervision, Funding acquisition, Project administration, Writing - review and editing; Jürgen Lademann, Conceptualization, Resources, Supervision, Funding acquisition, Methodology, Writing - review and editing; Marcus Maurer, Conceptualization, Data curation, Supervision, Funding acquisition, Methodology, Writing – original draft; Maxim E Darvin, Conceptualization, Data curation, Formal analysis, Supervision, Validation, Methodology, Writing – original draft, Project administration

### Author ORCIDs

Marius Kröger http://orcid.org/0000-0002-0148-4225
Maxim E Darvin http://orcid.org/0000-0003-1075-1994

### Ethics

Human subjects: Positive votes for the experiments have been obtained from the ethics committee of the Charité; - Universitätsmedizin Berlin (EA1/078/18, EA4/193/18, EA1/141/12), which were

conducted according to the Declaration of Helsinki (59th WMA General Assembly, Seoul, October 2008).

## Decision letter and Author response

Decision letter https://doi.org/10.7554/eLife.72819.sa1
Author response https://doi.org/10.7554/eLife.72819.sa2

## Additional files

### Supplementary files
• Transparent reporting form

### Data availability
The data have been deposited in Dryad.

The following dataset was generated:

| Author(s) | Year | Dataset title | Dataset URL | Database and Identifier |
|-----------|------|---------------|-------------|------------------------|
| Kröger M, Darvin M | 2021 | Macrophage FLIM raw data | https://dx.doi.org/10.5061/dryad.8gtht76q2 | Dryad Digital Repository, 10.5061/dryad.8gtht76q2 |

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

## Appendix 1

### FLIM data analysis

*Appendix 1—figure 1* shows the dependence of fluorescence decay parameters for macrophages on the integral fluorescence intensity.

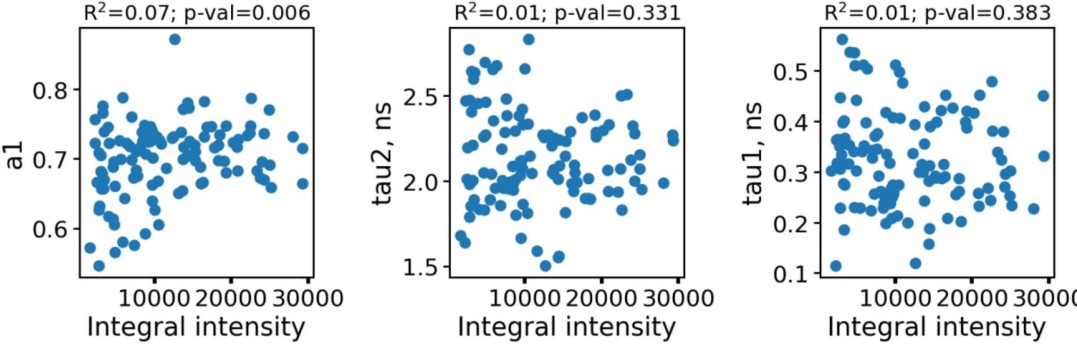

**Appendix 1—figure 1.** The dependence of fluorescence decay parameters for individual macrophages ($n$=110) on the integral fluorescence intensity (area under the fluorescence decay curve, upper row). As can be seen, the fluorescence decay parameters were independent on the intensity. To further confirm the absence of artefacts connected with parameters dependence on the FLIM data quality and processing algorithms, in **Appendix 1—figure 2** the dependence of fluorescence decay curves parameters on $\chi^2$ is shown.

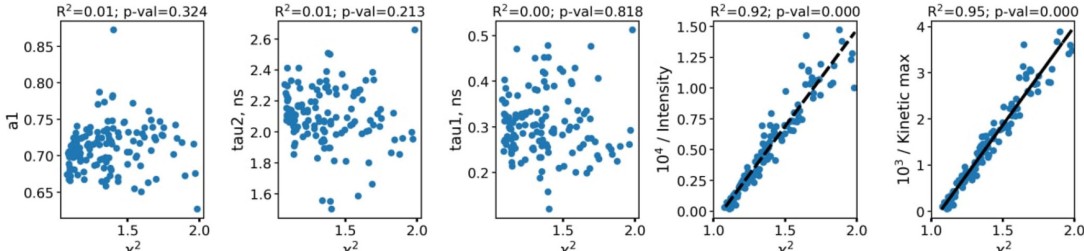

**Appendix 1—figure 2.** The dependence of fluorescence decay curves parameters on $\chi^2$. Each point corresponds to an individual macrophage cell ($n$=110). The dependence of $\chi^2$ on fluorescence intensity (both integral intensity per pixel and amplitude of the fluorescence decay curve, kinetic max) is a textbook knowledge—lower signal-to-noise ratio results in a worse fitting quality and higher $\chi^2$. Importantly, there was no correlation between the fluorescence decay parameters obtained from the decay curves and $\chi^2$; hence, there were no artifacts like lower fitting quality results in lower (or higher) values of fluorescence decay parameters. Summarizing, the fluorescence decay parameters obtained from bi-exponential fitting were independent on intensity (number of photons per pixel) and fitting quality, thus making it possible to use them as the descriptors for classification of cells. The comparison of fitting of the fluorescence decay for macrophages with two and three exponents ( **Appendix 1—figure 3**). As can be seen, an increase of the number of components does not result in an increase of the fitting quality.

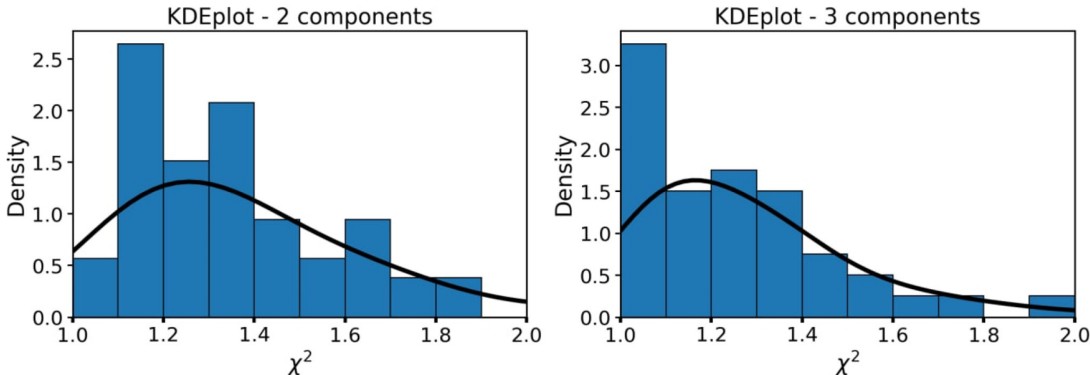

**Appendix 1—figure 3.** Distribution of the $\chi^2$ for the bi-exponential (left) model and three-exponential (right) decay models.

