## [Editor Report]

The authors have used measurements of endogenous fluorescence lifetimes in the two-photon stimulated NAD(P)H excitation-emission range to build an in vivo classifying for macrophage differentiation status in human dermis. The training data was derived from in vitro and ex vivo analysis of M1 and M2 polarised macrophages from peripheral blood, isolated from tissue or studies ex vivo in frozen sections with marker based validation. A machine learning approach for in vivo classification is presented and an approach to detect phagocytes in vivo is suggested.

---

## [Decision Letter]

**Decision letter after peer review:**

Thank you for submitting your article "Label-free imaging of macrophage phenotypes and phagocytic activity in the human dermis in vivo using two-photon excited FLIM" for consideration by *eLife*. Your article has been reviewed by 2 peer reviewers, including Michael L Dustin as Reviewing Editor and Reviewer #1, and the evaluation has been overseen by Aleksandra Walczak as the Senior Editor.

There are a number of positive aspects of your study including:

Using optical methods to non-invasively detect cells has significant interest for clinical and basic studies, and the impact of this study is considered high.

Identification of different FLIM signatures for macrophages polarized towards different phenotypes in vitro, and were able to compare these signatures to those of other cell types and macrophages in the skin.

They identified a few cells in the skin that expressed markers associated with macrophage polarization, and also exhibited the FLIM signatures that were established from the in vitro polarization studies.

At the same time, there are limitations of the study that can be addressed through the following Essential Revisions:

1) The description of M1 and M2 macrophages in the skin (in the introduction) is overly simplistic and the use of single markers (in Figure 3) to establish correlations is also simplistic. This should be updated and mixed phenotypes acknowledged.

2) It would be very useful to see how the classification tree fails ~10% of the time in relation to the plots in Figure S5. It's striking that the FLIM parameters generate a near perfect classification of M1 and M2 so it almost seems that adding information like cell shape may make it less accurate. Can the authors indicate in S5 which cells were mis-identified by the decision tree and if the reason is clear? If the FLIM parameters alone are fully discriminative it's not clear why the other parameters are helpful. Is there an "F-test" that can be done to assess the statistical value of each parameter that is added to the tree and if the improvement is greater then just adding another degree of freedom. Eventually this seems destined for some kind of experimental medicine application and this information would help determine where we are in terms of feasibility toward these near term goals. It would be important to report the number of in vivo tomograms that were done for each subject, the volume analyzed and amount of time to collect the tomograms.

3) Additional phenotype markers and evaluation of many more cells seems required to test the classifier. It's possible that the current 90% reliability is not actually statistically significant due to the low numbers. There are some significant technical concerns given that macrophages are a highly heterogeneous population of cells, particularly in vivo during an activation event such as injury. The few cells analyzed in Figure 3 are not sufficient given the heterogeneity of macrophages in vivo. Mixed phenotypes are common in vivo, and it is unclear how the FLIM signatures would change in response to such mixed signatures.

4) Visualizing a single phagocytosing cell in Figure 5 is also not sufficient to conclude that the method is capable of detecting phagocytosis events. Experiments in vitro treating macrophages with phagocytic targets need to be performed. An experiment in vivo would also strengthen the study, where phagocytic targets are applied to a skin wound. Controls without targets should be examined, along with quantitation across many cells. You may wish to focus on the classification task and perhaps only bring up phagocytosis as a possible confounder in this class-action process, but demonstrating the ability to detect phagocytosis in vivo may be another paper.

5) Is the lifetime signature stable over time? Can a single cell be imaged over time. This seems possible as macrophages are generally sessile. The study would be strengthened with further analysis that correlates FLIM signatures with metabolic state (free vs. bound NADPH). Could the lifetime relate to metabolic cycles in the issues that could change with time? This is a fundamental question that should be addressed in a first paper on this topic.

[Editors’ note: further revisions were suggested prior to acceptance, as described below.]

Thank you for resubmitting your work entitled "Label-free imaging of M1 and M2 macrophage phenotypes in the human dermis in vivo using two-photon excited FLIM" for further consideration by *eLife*. Your revised article has been evaluated by Aleksandra Walczak (Senior Editor) and a Reviewing Editor.

The manuscript has been improved but there are some remaining issues that need to be addressed, as outlined below:

This is a very interesting study that uses a label free method to classify macrophages in human dermis using two photon imaging. Additional data analysis and reporting is requested to fully document the very important work.

1. Justify your fitting model and include the decays pixel by pixel and also the fit profiles together with the goodness of the fit for all cases.

2. Compare with the Phasor plot, all your data.

3. Present all images with the corresponding number of photons acquired per pixel and plot these versus your mean lifetimes.

4. Calculate the true mean lifetime according to the canonical formula.

5. Present all your data in a pixel by pixel format.

*Reviewer #1 (Recommendations for the authors):*

Thank you for your clear response to the reviewer's concerns. You have addressed the issue of mixed phenotypes in healthy skin and your in vitro analysis of both in vitro differentiated and freshly isolated M1 and M2 like cells from skin show consistent two photon film signatures. You have provided information about the time involved in the measurements for each subject, the volume scanned and the number of cells found. You have also addressed the temporal stability in vivo consistent with the observation that the isolate cells maintain the flim phenotype even after isolation. The reporting on phagocytosis is now qualified and the limitations of the study are now appropriately acknowledged. I have no further recommendations.

*Reviewer #3 (Recommendations for the authors):*

In general the approach and validation of the technique is appropriate. I would be interested though to check the relative number of photons collected per pixel versus the different lifetimes to make sure that all lifetimes collected and analyzed are independent on the intensity/number of photons.

Also, some of the lifetimes recovered were very short and close to the resolution of the instrument response function (IRF); which by the way seems not to be recovered experimentally but rather produced by the SPCI software. Please, clarify and also if it was not recovered experimentally produce these data.

This brings me to my next point. The authors employ a double exponential approach and present both the average lifetime (calculated based on tau1 and tau2 and not with the appropriate formula, see Padilla-Parra et al., 2008 Biophys J). The authors do not show the fluorescence decays for each cell, together with the double exponential fits nor the goodness of these fits or the Χ2 (chi square). This data is fundamental to understanding if all FLIM data recovered can be fitted to the double exponential model. Also, the authors assume a double exponential approach but they do not justify their choice. Which are the two populations you are assuming to co-exist in the cells/dermis? Are you always measuring NADPH/NADP? In this case, have you shown the value of these two lifetimes in vitro, and then you should fix these two lifetimes and recover different proportions, right? The model to fit your data should be discussed and justified. I assume that if you use a triple exponential most of your stats will be better. Please discuss this also in the context of the number of photons.

Also, all the parameters recovered from the fit should be also shown pixel by pixel so that we can understand how these data vary as a function of the number of photons, or the error (Chi Square).

In Figure 4 the authors decide to show the phasor plot (Digman et al., 2008 Biophys J). This is a nice approach that does not necessitate the assumption of a model. You should present all data comparing this approach and the fitting approach utilizing a double exponential and also discuss how your data varies depending on the number of photons. It is possible that some lifetime distributions arise from the fact that in some cells the high intensity values give longer lifetimes and in others where there is a lower signal to noise, lower lifetimes are observed. When plotting your lifetimes versus the number of photons pixel by pixel no dependence should be observed. This will help to validate your data.

---

## [Author Response]

At the same time, there are limitations of the study that can be addressed through the following Essential Revisions:1) The description of M1 and M2 macrophages in the skin (in the introduction) is overly simplistic and the use of single markers (in Figure 3) to establish correlations is also simplistic. This should be updated and mixed phenotypes acknowledged.

We thank the reviewer for this important comment. We agree that the M1/M2 is simplified and it is clear that macrophage phenotype diversity goes beyond that. Not only differ macrophages with regards to their origin but they can adopt to various responsive phenotypes with shared and mixed signatures, depending on the stimulus they experience. This being said, macrophage polarization requires certain stimuli, which might not be present in healthy skin. In literature, spontaneously mixed phenotypes in healthy skin have not been found.

The precise role of skin ΜΦs and their M1 and M2 phenotypes in health and disease remains to be elucidated. In skin disease such as melanoma (Bardi GT, Smith MA, Hood JL. Melanoma exosomes promote mixed M1 and M2 macrophage polarization. Cytokine. 105: 63-72, 2018), systemic sclerosis (Trombetta AC, Soldano S, Contini P, et al. A circulating cell population showing both M1 and M2 monocyte/macrophage surface markers characterizes systemic sclerosis patients with lung involvement. Respir Res. 19(1): 186, 2018), Lupus (Chong BF, Tseng LC, Hosler GA, et al. A subset of CD163+ macrophages displays mixed polarizations in discoid lupus skin. Arthritis Res Ther. 17: 324, 2015) and Lipopolysaccharide Tolerance (O'Carroll C, Fagan A, Shanahan F, Carmody RJ. Identification of a unique hybrid macrophage-polarization state following recovery from lipopolysaccharide tolerance. J Immunol. 192(1): 427-436, 2014), ΜΦs polarizations leading to mixed M1 and M2 phenotypes can be observed.

Moreover, this study did not aim to fully characterize the FLIM signatures of all possible macrophage subpopulations but to describe a method to visualize macrophages in the healthy skin without the need of external labeling. Further studies are definitely required to investigate mixed phenotypes in health and skin disease and disorders.

The additional information is updated and acknowledged in the main text on page 1: line 46 and page 13: line 454.

2) It would be very useful to see how the classification tree fails ~10% of the time in relation to the plots in Figure S5. It's striking that the FLIM parameters generate a near perfect classification of M1 and M2 so it almost seems that adding information like cell shape may make it less accurate. Can the authors indicate in S5 which cells were mis-identified by the decision tree and if the reason is clear? If the FLIM parameters alone are fully discriminative it's not clear why the other parameters are helpful. Is there an "F-test" that can be done to assess the statistical value of each parameter that is added to the tree and if the improvement is greater then just adding another degree of freedom. Eventually this seems destined for some kind of experimental medicine application and this information would help determine where we are in terms of feasibility toward these near term goals. It would be important to report the number of in vivo tomograms that were done for each subject, the volume analyzed and amount of time to collect the tomograms.

We thank the reviewer for this important recommendation. The high accuracy for M1 macrophages is owed to the fact that M1 macrophages have the shortest fluorescence lifetimes *τ_1_* and *τ_2_* and the highest ratio of *a_1_*/*a_2_* of the cells in the model. M2 macrophages can be confused in rare cases with resting mast cells or/and dendritic cells in vitro due to the overlap in *τ_2_* fluorescence lifetime. These TPE-FLIM parameters can be seen in Table 1 and the phasor plot in Figure 4. The failure is also visible in the 2D segmentation of isolated dermal macrophages where we find areas located closely together, which can be interpreted as superposition of parameters.

In the figures we show the cells (area marked red) which were misidentified by the decision tree in our in vitro and in vivo experiments, in Figure 2—figure supplement 3 and Figure 4—figure supplement 1 (new numbering).

Figure S3 and Figure S5 have been replaced by the new (Figure 2—figure supplement 3 and Figure 4—figure supplement 1 in a new numbering) in the supplementary materials showing additionally cells misidentified by the decision tree.

The misidentification was due to the different values of TPE-FLIM parameters and the proximity to the classification conditions.

As a reminder the independent TPE-FLIM parameters are *a*_1_, *a*_2_, *τ*_1_, *τ*_2_, the dependent TPE-FLIM variables *τ*_m_, *τ*_2_/*τ*_1_, *a*_1_/*a*_2_, (*a*_1_-*a*_2_)/(*a*_1_+*a*_2_) and the fluorescence intensity.

With 10,000 repetitions of random training data and test data splits, we are sure the model is not overfitted to the data. We have to separate between true positive rate and true negative rate, meaning the difference between the ability of the model to predict macrophages as macrophages (true positive rate, TPR) and to predict other dermal cells as non-macrophages (true negative rate, TNR). The TPR for M2 macrophages is lower compared to M1 macrophages due to a more common FLIM signature, where less M2 macrophages get classified as M2 macrophages. Yet, the TNR is higher for M2 macrophages compared to M1 macrophages.

A “F-Test” is not needed and not applicable in the decision tree model, because the significance of the parameter is directly tied to the use in the decision tree model. As can be seen in the decision tree model in Figure 4—figure supplement 5 (new numbering), the best classification is done with TPE-FLIM parameters only, shape and fluorescence intensity wasn’t used in this model, thus the degree of freedom is limited by the relevant parameter in the decision tree model. The 8 TPE-FLIM parameters: *τ*_1_, *τ*_2_, *τ*_m_*, a*_1_, *a*_2_, *τ*_2_/*τ*_1_, *a*_1_/*a*_2_, (*a*_1_-*a*_2_)/(*a*_1_+*a*_2_), obtained after bi-exponential approximation of the decay curve are enough for a meaningful classification. The fluorescence intensity will be a more important feature when only in vivo cells in the papillary and reticular dermis will are classified and the fluorescence intensity is comparable independent of absorption and scattering effects, due to the same depth in the skin. The cell shape and fluorescence intensity are promising indicators to separate functionally different macrophage phenotypes in a comprehensive classification model with more dermal cells in the future. The parameters used in the model are described again in the Results section on page 10 and discussed from line 286 and in the Discussion section on page 14: line 484.

“Six to twelve in vivo tomograms (different skin areas) were measured per subject, the investigated volume is from ≈70 µm depth in the papillary dermis to ≈130 µm depth in the reticular dermis with an image size of 150 µm. This adds up to (60×150×150) µm³ times six to twelve images, with a total volume of ≈0.0081 to 0.0162 mm³ of papillary and reticular dermis seen per subject, the average time spent was ≈30 minutes per subject and the acquisition time was 6.8 s per image.”

This information has been added to the methods section of main text on page 15: line 558.

3) Additional phenotype markers and evaluation of many more cells seems required to test the classifier. It's possible that the current 90% reliability is not actually statistically significant due to the low numbers. There are some significant technical concerns given that macrophages are a highly heterogeneous population of cells, particularly in vivo during an activation event such as injury. The few cells analyzed in Figure 3 are not sufficient given the heterogeneity of macrophages in vivo. Mixed phenotypes are common in vivo, and it is unclear how the FLIM signatures would change in response to such mixed signatures.

It is desirable that more cells have to be tested in the future to further strengthen the model and classification accuracy. In this pilot study all measures were met to avoid the overfitting of the data or the statistical insignificance of the classification method due to low numbers. First, the training data set and the test set was split randomly 10,000 times and the model was tested on it, this approach removes the possibility of random spikes in sensitivity and specificity and ensures a sufficient size of training and test sets. The standard deviation of both the specificity and sensitivity was reasonably low, indicating no irregularities. Additionally, a k-fold cross-validation with k=5 was executed to avoid unevenly distributed data, with the results: (0.87; 0.92; 0.87; 0.89; 0.94) and mean of k-fold scores using the cross_val_score method is 0.90. The macrophages misidentified by the decision tree (answer to the first reviewer’s question) and now shown in Figure 2—figure supplement 3 and Figure 4—figure supplement 1 (new numbering), could potentially be macrophages of a mixed phenotype, however this conclusion was not proved in this study and needs additional investigations. The main text is updated on page 10: line 296 in the Results section and on page 14: line 484 in the Discussion section.

Regarding the heterogeneity of macrophages please also refer to the response to the question 1, all subjects were investigated on healthy and asymptomatic skin areas without pre-existing skin conditions. It is unclear whether and to what amount the mixed phenotypes occurring in disease are existent in healthy and asymptomatic skin.

Hence, we focused on M1 and M2 phenotypes in this study, that is a simplified yet realistic model, and mixed phenotypes will be a subject of further investigations.

4) Visualizing a single phagocytosing cell in Figure 5 is also not sufficient to conclude that the method is capable of detecting phagocytosis events. Experiments in vitro treating macrophages with phagocytic targets need to be performed. An experiment in vivo would also strengthen the study, where phagocytic targets are applied to a skin wound. Controls without targets should be examined, along with quantitation across many cells. You may wish to focus on the classification task and perhaps only bring up phagocytosis as a possible confounder in this class-action process, but demonstrating the ability to detect phagocytosis in vivo may be another paper.

We thank the reviewer for the insightful comment and suggestion. In this paper we have focused on the classification task as it arises from the questions above with added k-fold cross validation and extended discussion. We have changed the title to “Label-free imaging of M1 and M2 macrophage phenotypes in the human dermis in vivo using two-photon excited FLIM”. Figure 5 has been moved to the supplementary information, see Figure 4—figure supplement 4 (new numbering).

The authors agree with the reviewer that in vitro and in vivo study of phagocytosis is required. However, the authors decided to adhere to these pilot results about visualization of phagocytosing macrophages in the skin in the manuscript due to the following reasons.

As a further evidence, the erythrophagocyting cells, presented by our group (Yakimov et al., Label-free characterization of white blood cells using fluorescence lifetime imaging and flow-cytometry: molecular heterogeneity and erythrophagocytosis, Biomedical Optics Express. 10(8): 4220-4236, 2019) exhibit strong shortening of fluorescence decay, resembling the cell shown in our study. As a defence mechanism phagocytosing ΜΦs reduce the acidify, their adjacent extracellular environment and simultaneously ROS are produced, which acts as bactericide and is known to decrease fluorescence lifetimes (Haka AS, Grosheva I, Chiang E, et al. Macrophages create an acidic extracellular hydrolytic compartment to digest aggregated lipoproteins. Mol Biol Cell. 20(23): 4932-4940, 2009; Teixeira J, Basit F, Swarts HG, et al. Extracellular acidification induces ROS- and mPTP-mediated death in HEK293 cells. Redox Biol. 15: 394-404, 2018; Slauch JM. How does the oxidative burst of macrophages kill bacteria? Still an open question. Mol Microbiol. 80(3): 580-583, 2011; Li, W., Houston, K. and Houston, J. Shifts in the fluorescence lifetime of EGFP during bacterial phagocytosis measured by phase-sensitive flow cytometry. Sci Rep 7: 40341, 2017). Thus, the observed cells in the skin and their TPE-FLIM parameters and morphology are in agreement with previous results and knowledge about phagocytosing macrophages.

We modified the paragraph “TPE-FLIM can distinguish resting from phagocytosing human skin M1 ΜΦs in vivo“, please see page 9: line 266 and the discussion on page 13: line 467.

5) Is the lifetime signature stable over time? Can a single cell be imaged over time. This seems possible as macrophages are generally sessile. The study would be strengthened with further analysis that correlates FLIM signatures with metabolic state (free vs. bound NADPH). Could the lifetime relate to metabolic cycles in the issues that could change with time? This is a fundamental question that should be addressed in a first paper on this topic.

We thank the reviewer for the interesting technical question. The fluorescence lifetime of in vitro cells is stable for at least one hour and lies in the range of the variation in lifetime presented in Table 1. The in vivo cells are stable over the course of the measurement and the stability of their TPE-FLIM parameters is also within the standard deviation. One indicator of the lifetime stability is that ex vivo experiments agree with the other measurements.

A single cell could be imaged over time, however the in vivo subjects cannot be measured for longer than half an hour due to human restraints.

We performed a time series images of macrophages isolated from periocular skin during 60 minutes acquired in vitro every 5 minutes at 6.8 s scanning time at 6 mW, this is a total of 81.6 s of scanning at 80 MHz repetition rate. The TPE-FLIM images for a mixture of M1 and M2 macrophages is presented for two time points: 0 and after 60 minutes. As can be seen, TPE-FLIM parameters of M1 and M2 macrophages are stable and vary within the corresponding standard deviation presented in Table 1. These TPE-FLIM images are added as Figure 2—figure supplement 4 in the paper.

We performed a time series of dermal M1 and M2 macrophage images during 30 minutes acquired in vivo every minute at 6.8 s scanning time and at 50 mW, this is a total of 210.8 s of scanning at 80 MHz repetition rate. The TPE-FLIM image for M1 and M2 macrophages in vivo is presented in the figure for seven time points: 0, 5, 10, 15, 20, 25 and 30 minutes. As can be seen, TPE-FLIM parameters for recorded M1 and M2 macrophages are stable and vary within the standard deviation presented in Table 1. These TPE-FLIM images are added as Figure 4—figure supplement 2,3 (new numbering) in the paper.

No significant changes can be observed in TPE-FLIM parameters in vitro and in vivo during the stability measurements over period of 30 to 60 min.

In the literature macrophage infiltration is measured in a matter of days by Baeten *et al.* (Baeten, Kurt, et al. Visualisation of the kinetics of macrophage infiltration during experimental autoimmune encephalomyelitis by magnetic resonance imaging. Journal of Neuroimmunology, 195:1-6, 2008) and Lauterbach *et al.* (Lauterbach, Mario A., et al. Toll-like receptor signaling rewires macrophage metabolism and promotes histone acetylation via ATP-citrate lyase. Immunity, 51: 997-1011, 2019) showed change in metabolism by toll-like receptors over the course of hours with a maximum at 4 hours. In the time frame of the measurements of our pilot study we don’t expect significant changes in fluorescence lifetime and only snapshots of the metabolic status of the in vivo macrophage can be taken.

The Discussion section has been completed with these results based on review’s suggestion, please see page 11: line 333.

Furthermore, changes in TPE-FLIM parameter can generally be induced by photo bleaching, changes in temperature, cellular metabolic changes and photoproduct formation.

The effect of photo bleaching is not present in this work. As it has been shown in the literature (L. Tiede and M. Nichols, Photobleaching of Reduced Nicotinamide Adenine Dinucleotide and the Development of Highly Fluorescent Lesions in Rat Basophilic Leukemia Cells During Multiphoton Microscopy, Photochemistry and Photobiology, 82: 656-664, 2006), no photo bleaching effects have been observed in cells up to 5 min of data collection using excitation power of 7 mW at 80 MHz repetition rate, 740 nm wavelength and an objective with NA of 1.3. Since in our case the acquisition time was 6.8 s per image and below 4 minutes in total at lower excitation power in vitro and due to absorption and scattering lower excitation power in vivo, defined by emission photons per mW, we assure that no photo bleaching occurred during the fluorescence measurements of NAD(P)H. The overall photon counting rate did not decrease during measurements, which is an indication of absence of any TPE-induced photo bleaching effects.

It cannot exclude the influence of excitation light on metabolism even in the absence of photo bleaching. To assess if any changes have been triggered upon light illumination, the changes in the level of DNA synthesis must be evaluated. This has been done, for example, in the work (N. Nichols et al., Reduction in DNA synthesis during two-photon microscopy of intrinsic reduced nicotinamide adenine dinucleotide fluorescence, Photochemistry and Photobiology, 81: 259-269, 2005). Evaluation of the data shows that we are working below the threshold doses that may trigger a reduction in DNA synthesis.

It is known that both fluorescence intensity and fluorescence lifetime of NAD(P)H depend on temperature. In general, an increase in temperature is accompanied by decreases in fluorescence lifetime and quantum yield due to increased efficiency of non-radiative processes related to thermal effects, such as collisions with solvent molecules, intramolecular vibrations, and rotations. We do not see a reduction in the fluorescence intensity during the experiments, therefore we believe that at the intensity levels we used, the thermal effects are negligible. In the skin in vivo the temperature is regulated by the transport of water, blood and lymph, and in vitro in the cell culture, the temperature is stable at room temperature for the whole measurement.

In recent research (Shirmanova, M.V., et al. "FUCCI-Red: a single-color cell cycle indicator for fluorescence lifetime imaging." Cellular and Molecular Life Sciences 78: 3467-3476, 2021), the correlation between the cell cycle and fluorescence lifetime of NAD(P)H has been examined and showed that proliferating cells have shorter mean lifetime of NAD(P)H, an indicator of glycolytic shift.

It is known that photo oxidation induces a broadband excitation spectrum and red edge excited fluorescence related to oxidation products of aromatic amino acids from an irradiation of 10 mW/cm intensity during 6 h at 254 nm (Semenov AN, Yakimov BP, Rubekina AA, et al. The Oxidation-Induced Autofluorescence Hypothesis: Red Edge Excitation and Implications for Metabolic Imaging. Molecules. 25(8): 1863, 2020), leading to shortened fluorescent lifetimes and an increase in fluorescence intensity. In our investigations given we used two-photon excitation at 760 nm and significantly shorter irradiation time, it can be excluded that photoproducts had an impact on the measured TPE-FLIM parameters in vitro and in vivo.

Regarding the ΜΦ polarization, the paradigm shifts towards a less strict classification compared to M1 (IFN-γ/LPS-polarized) and M2 (IL-4-polarized) (Murray PJ. Macrophage Polarization. Annu Rev Physiol. 79: 541-566, 2017). While this categorization is useful in clinical terms, the multitude of parameters leading to the differentiation process leaves ΜΦs with wide-ranging properties both in expression of markers, appearance and TPE-FLIM parameters.

M1 (IFN-γ/LPS-polarized) ΜΦs rely on the NADH oxidase and production of ROS, which is shown by fluorescent lifetimes of under 250 ps and mitochondrial fission, which can indicate the bright spots, whereas M2 (IL-4-polarized) rely on oxidative phosphorylation and fatty acid oxidation, together with mitochondrial fusion, it can explain the homogeneous appearance of M2 ΜΦs (Xu Q, Choksi S, Qu J, et al. NADPH Oxidases Are Essential for Macrophage Differentiation. J Biol Chem. 291(38): 20030-20041, 2016; Ramond E, Jamet A, Coureuil M and Charbit A. Pivotal Role of Mitochondria in Macrophage Response to Bacterial Pathogens. Front. Immunol. 10:2461, 2019; Swindle EJ, Hunt JA, Coleman JW. A comparison of reactive oxygen species generation by rat peritoneal macrophages and mast cells using the highly sensitive real-time chemiluminescent probe pholasin: inhibition of antigen-induced mast cell degranulation by macrophage-derived hydrogen peroxide. J Immunol. 169(10): 5866-5873, 2002).

Future investigations will address the change in TPE-FLIM parameter with metabolic stress and shifts in macrophage polarization.

Additional information was added in the Results section on page 5: line 169, page 8: line 245 and in the Discussion section on page 10: line 320, page 11: line 333 and page 13: line 440.

[Editors’ note: further revisions were suggested prior to acceptance, as described below.]

Reviewer #1 (Recommendations for the authors):Thank you for your clear response to the reviewer's concerns. You have addressed the issue of mixed phenotypes in healthy skin and your in vitro analysis of both in vitro differentiated and freshly isolated M1 and M2 like cells from skin show consistent two photon film signatures. You have provided information about the time involved in the measurements for each subject, the volume scanned and the number of cells found. You have also addressed the temporal stability in vivo consistent with the observation that the isolate cells maintain the flim phenotype even after isolation. The reporting on phagocytosis is now qualified and the limitations of the study are now appropriately acknowledged. I have no further recommendations.

We are grateful to the Reviewer for the positive assessment of our work and responses.

Reviewer #3 (Recommendations for the authors):In general the approach and validation of the technique is appropriate. I would be interested though to check the relative number of photons collected per pixel versus the different lifetimes to make sure that all lifetimes collected and analyzed are independent on the intensity/number of photons.

We agree with the Reviewer that in the FLIM data correlation between the fluorescence intensity (e.g. photon counts per pixel) and decay parameters could be observed. The question of the dependence of fluorescence decay parameters on the intensity is also connected with further questions of the Reviewer, namely, the dependence of fitting quality on the intensity. To address this issue, Appendix 1-figure 1 shows the dependence of fluorescence decay parameters for macrophages on the integral fluorescence intensity.

As can be seen, the fluorescence decay parameters were independent on the intensity (also note that here a1 is the normalized fraction of the first component, i.e., a_1_ + a_2_ = 1).

We also note that all the fluorescence decay parameters presented in the paper were calculated for individual cells, i.e., averaged over ~100 pixels with binning 3 (i.e., summation over 48 neighbouring pixels was performed), thus providing a reasonable number of photons for fitting. We would also like to add that the question of FLIM analysis over individual cells (averaging, the dependence of fitting errors on the number of photons etc) was discussed in our recent paper (Shirshin et al., PNAS, 2022, 10.1073/pnas.2118241119).

To further confirm the absence of artefacts connected with parameters dependence on the FLIM data quality and processing algorithms, in Appendix 1-figure 2 the dependence of fluorescence decay curves parameters on χ^2^ is shown.

The dependence of χ^2^ on fluorescence intensity (both integral intensity per pixel and amplitude of the fluorescence decay curve, “kinetic max”) is a textbook knowledge – lower signal to noise ratio results in a worse fitting quality and higher χ^2^ (please see two panels to the right in Appendix 1-figure 2). Importantly, there was no correlation between the fluorescence decay parameters obtained from the decay curves and χ^2^, hence, there were no artefacts like lower fitting quality resulting in lower (or higher) values of fluorescence decay parameters.

Summarizing, the fluorescence decay parameters obtained from biexponential fitting were independent on the intensity (number of photons per pixel) and fitting quality, thus making it possible to use them as the descriptors for classification of cells.

This discussion was added to the SI and briefly addressed in the Materials and methods section of the paper.

Also, some of the lifetimes recovered were very short and close to the resolution of the instrument response function (IRF); which by the way seems not to be recovered experimentally but rather produced by the SPCI software. Please, clarify and also if it was not recovered experimentally produce these data.

Indeed, for some cells the fluorescence decay was quite fast. The IRF of our setup was measured using the SHG signal and its FWHM was estimated as ~100 ps (please see Author response image 1).

**Author response image 1. sa2fig1:** Fluorescence decay curves for individual macrophages, from which the values of τ_1_ < 300 ps (left) and τ_1_ > 300 ps (right) were obtained using the biexponential decay model. The black curve corresponds to IRF of the setup. The curves were normalized to the maximum.

In Author response image 1, fluorescence decay curves for individual cells are divided into two classes: those with τ_1_ < 300 ps (left) and τ_1_ > 300 ps (right). It can be seen that even in the case of short lifetimes, the IRF is more “narrow” than the fluorescence decay curve. We definitely agree with the Reviewer that some of the lifetimes are too short, however, (1) the absence of intercorrelation between the fluorescence decay parameters and (2) the fact that for the majority of curves τ_1_ significantly exceeds the FWHM of the IRF prove that the decay parameters recovered by the procedure employed in our work can used can be used for classification of cells into subpopulations.

This brings me to my next point. The authors employ a double exponential approach and present both the average lifetime (calculated based on tau1 and tau2 and not with the appropriate formula, see Padilla-Parra et al., 2008 Biophys J). The authors do not show the fluorescence decays for each cell, together with the double exponential fits nor the goodness of these fits or the Χ2 (chi square). This data is fundamental to understanding if all FLIM data recovered can be fitted to the double exponential model. Also, the authors assume a double exponential approach but they do not justify their choice. Which are the two populations you are assuming to co-exist in the cells/dermis? Are you always measuring NADPH/NADP? In this case, have you shown the value of these two lifetimes in vitro, and then you should fix these two lifetimes and recover different proportions, right? The model to fit your data should be discussed and justified. I assume that if you use a triple exponential most of your stats will be better. Please discuss this also in the context of the number of photons.

This question of the Reviewer is connected with the experimental basis underlying the metabolic FLIM of NADH. Below, we’ll briefly justify our model based on the literature knowledge base and understanding of the NAD(P)H fluorescence.

It is commonly accepted that the fluorescence signal of a cell at 700–760 nm excitation can be ascribed to NAD(P)H (note that NAD+ and NADP+ do not fluoresce). The idea of using FLIM for metabolic imaging is based on the assumption that the first (fast) component in fluorescence decay of NAD(P)H is originated form free NAD(P)H and the second (slow) component is originated from NAD(P)H bound to enzymes. The reasons of the differences in lifetimes of free and bound NAD(P)H forms are extensively studied (Cao et al. Chem Phys Lett, 726: 18, 2019; Cao et al. J Phys Chem B 124: 6721, 2020; Gorbunova et al. J Phys Chem B 125: 9692, 2021). The basis of using the a1/a2 ratio in the fluorescence decay of NAD(P)H for assessing metabolic (redox) alterations in cells is reviewed in e.g. (Blacker et al. Free Radic Biol Med 100: 53, 2016; Kolenc et al. Antiox Redox Signal 30: 875, 2019).

The lifetimes of NAD(P)H can not be fixed during fitting due to the following reasons. Firstly, NAD(P)H in cells is bound to a manyfold of enzymes, and different enzymes cause different lifetimes of the bound NAD(P)H (in the 1.5–6 ns range Blacker et al. Free Radic Biol Med 100: 53, 2016; Kolenc et al. Antiox Redox Signal 30: 875, 2019; Shirshin et al. Biochem 84: 69, 2019). Hence, the second lifetime of NAD(P)H represents the average over different bound NAD(P)H molecules, and its variation is meaningful and corresponds to redistribution of NAD(P)H between distinct proteins. Secondly, free NAD(P)H in aqueous solution exhibits two lifetimes, ca. 200 and 800 ps (Cao et al. Chem Phys Lett, 726: 18, 2019; Cao et al. J Phys Chem B 124: 6721, 2020; Gorbunova et al. J Phys Chem B 125: 9692, 2021). The origin of these components is debatable and originates either from conformational states of the whole molecule or from isomerisation in the nicotinamide ring (Gorbunova et al. Phys Chem Chem Phys 22: 18155, 2020; Gorbunova et al. J Phys Chem B 125: 9692, 2021). The latter lifetime is close to that for some bound NAD(P)H forms (Vishwasrao et al. J Biol Chem 280: 25119, 2005). Hence, the mechanism of NAD(P)H decay in such a complex environment as a cell is complicated, and for metabolic FLIM a simple two-component model is used (see, e.g., the Becker and Hickl guideline https://www.becker-hickl.com/applications/metabolic-imaging/). Summarizing, the biexponential fitting procedure used in our work is a standard, and it is also used in the papers on macrophages (Lemire et al. Int J Mol Sci 23: 2338, 2022; Alfonso-García et al. J Biomed Opt 21: 46005, 2016).

We would also like to note that in the work of (Blacker et al. Nat Comm 5: 1, 2014) an approach to separate NADH and NADPH impacts using three exponentials was suggested, but it is not a standard and requires high intensity signals.

According to the theoretical estimations, if the number of photons is less than 10000, there is no need to use more than two exponentials (Kollner et al. Chem Phys Lett 200: 199204, 1992). As for the endogeneous fluorescence signal is rather weak (ca 1000 photons in the fluorescence decay curve maximum), biexponential approximation is appropriate, as stated above. To additionally illustrate this fact, we performed comparison of fitting of the fluorescence decay for macrophages with 2 and 3 exponents (Appendix 1-figure 3). As can be seen, an increase of the number of components does not result in a notable increase of the fitting quality.

Also, all the parameters recovered from the fit should be also shown pixel by pixel so that we can understand how these data vary as a function of the number of photons, or the error (Chi Square).

Please see the answer to the first question.

In Figure 4 the authors decide to show the phasor plot (Digman et al., 2008 Biophys J). This is a nice approach that does not necessitate the assumption of a model. You should present all data comparing this approach and the fitting approach utilizing a double exponential and also discuss how your data varies depending on the number of photons. It is possible that some lifetime distributions arise from the fact that in some cells the high intensity values give longer lifetimes and in others where there is a lower signal to noise, lower lifetimes are observed. When plotting your lifetimes versus the number of photons pixel by pixel no dependence should be observed. This will help to validate your data.

The absence of intensity dependence of fluorescence decay parameters was shown (please see Appendix 1-figure 1). Below we will address the correlations between the phasor approach and biexponential fitting.

The phasor approach is a fit-free method used to represent the FLIM data. The basics of the phasor plot is that the position of the point on the phasor plane represents the decay parameters of the fluorescence curve. We’ve performed a detailed analysis of the capabilities of phasor plot, biexponential fitting and K-means clustering in the analysis of the NAD(P)H FLIM data (similar to the presented paper) in our recent work (Shirshin et al. PNAS, 119: e2118241119, 2022). Briefly, the results of the biexponential fitting and phasor analysis are almost similar, with fitting being more beneficial in separating between cells subpopulations with close fluorescence lifetimes. Hence, these two approaches generally yield similar information when aiming at detecting cells subpopulations.

We also demonstrate the correlation between the phasor data (c and s) and the fluorescence decay parameters obtained from biexponential fitting for single macrophage cells in Author response image 2.

**Author response image 2. sa2fig2:** Correlations between the phasor plot parameters (c and s) and fluorescence decay parameters obtained from the biexponential fitting. Each point corresponds to an individual macrophage.

As can be seen, the c value (cosine, x-axis of the phasor plot) is correlated with all decay parameters, while s (sine, y-axis) mainly correlates with a1. We do not draw out any conclusions from these correlations, as the very idea of the phasor plot is the representation of similar data compared to the fitting procedure, hence, the parameters must be correlated (and it is so).